# Quantifying spectral information about source separation in multisource odour plumes

**Sina Tootoonian**[1]*, **Aaron C. True**[2], **Elle Stark**[2], **John P. Crimaldi**[2], **Andreas T. Schaefer**[1,3]

**1** Sensory Circuits and Neurotechnology Laboratory, The Francis Crick Institute, London, United Kingdom, **2** Department of Civil, Environmental and Architectural Engineering, University of Colorado, Boulder, CO, United States of America, **3** Department of Neuroscience, Physiology and Pharmacology, University College London, United Kingdom

* sina.tootoonian@crick.ac.uk

**Data Availability Statement:** Data for this work is available on Figshare at 10.25418/crick.26494648.

**Funding:** This work was supported by the Francis Crick Institute (https://www.crick.ac.uk), which receives its core funding from Cancer Research

## Abstract

Odours released by objects in natural environments can contain information about their spatial locations. In particular, the correlation of odour concentration timeseries produced by two spatially separated sources contains information about the distance between the sources. For example, mice are able to distinguish correlated and anti-correlated odour fluctuations at frequencies up to 40 Hz, while insect olfactory receptor neurons can resolve fluctuations exceeding 100 Hz. Can this high-frequency acuity support odour source localization? Here we answer this question by quantifying the spatial information about source separation contained in the spectral constituents of correlations. We used computational fluid dynamics simulations of multisource plumes in two-dimensional chaotic flow environments to generate temporally complex, covarying odour concentration fields. By relating the correlation of these fields to the spectral decompositions of the associated odour concentration timeseries, and making simplifying assumptions about the statistics of these decompositions, we derived analytic expressions for the Fisher information contained in the spectral components of the correlations about source separation. We computed the Fisher information for a broad range of frequencies and source separations for three different source arrangements and found that high frequencies were more informative than low frequencies when sources were close relative to the sizes of the large eddies in the flow. We observed a qualitatively similar effect in an independent set of simulations with different geometry, but not for surrogate data with a similar power spectrum to our simulations but in which all frequencies were *a priori* equally informative. Our work suggests that the high-frequency acuity of olfactory systems may support high-resolution spatial localization of odour sources. We also provide a model of the distribution of the spectral components of correlations that is accurate over a broad range of frequencies and source separations. More broadly, our work establishes an approach for the quantification of the spatial information in odour concentration timeseries.

United Kingdom (https://www.cancerresearchuk.org, Grant FC001153); the UK Medical Research Council (https://www.ukri.org/councils/mrc/, Grant FC001153); the Wellcome Trust (https://wellcome.org, Grant FC001153); Wellcome Trust Investigator Grant 110174/Z/15/Z (to ATS); and the National Science Foundation (https://www.nsf.org)/Canadian Institutes of Health Research (https://cihr-irsc.gc.ca)/German Research Foundation (https://www.dfg.de)/Fonds de Recherche du Québec (https://frq.gouv.qc.ca)/UK Research and Innovation–Medical Research Council Next Generation Networks for Neuroscience Program (Award No. 2014217 to JPC and ATS). The funders had no role in study design, data collection and analysis, decision to publish, or preparation of the manuscript.

**Competing interests:** The authors have declared that no competing interests exist.

# 1 Introduction

Olfactory signals enable a multitude of crucial behaviors observed in terrestrial organisms, from male moths navigating to females from hundreds of meters away [1, 2], to mice detecting and avoiding predators [3], to honeybees foraging for nectar in a flower among a field of vegetation [4]. Such feats of olfactory navigation pose complex problems, due in part to a turbulent transport process that causes odour signals to become patchy and highly fluctuating away from the source [5]. Even in a single-source context, strategies for olfactory search span a range of complexity [6], including reactive strategies in which a searcher reorients upwind upon odour detection [7, 8], information-theoretic approaches that exploit the spatiotemporal dynamics of odour signals [9–11], and strategies that incorporate non-olfactory cues in the search (such as visual cues as in [12], public cues from other searchers as in [13], or direction of motion of odor as in [14]). Recent work employs machine learning, including neural networks and reinforcement learning, to discover data-driven navigational strategies from large training sets [15].

Such olfactory search strategies are often studied in the context of single-source searches, but in naturalistic odour plumes, odour signals do not arrive in isolation but instead arrive in a fluctuating mixture of multiple odorants, further complicating navigation tasks [16]. In such multi-source contexts, a navigating animal must both identify the odour(s) of interest and localize the target odour(s). The relative distance between two target odours may be of particular interest to an organism; for example, the distance of a predator to a food source or the proximity of multiple food sources to each other. As noted early by Hopfield [17], the modulation of correlation by intersource distance can help animals locate odour sources in the environment, i.e. perform olfactory scene segmentation. To resolve such correlations requires an ability to parse signals at high frequency. Insects indeed demonstrate the capacity to distinguish high-bandwidth odor signals at frequencies up to 100 Hz [18], and in a multi-source context, moths have even been shown to discriminate odor sources separated by as little as 1 mm in space or 0.001 s in time [19]. Mammalian olfaction has generally been considered a 'slow' modality, but recent research demonstrated the ability of mice to discriminate fast odour fluctuations at frequencies of up to 40 Hz, much higher than expected [20].

Given the sensitivity of animals to high-bandwidth information in odour signals, can the correlations between two odour signals theoretically be exploited to estimate the distance between the two sources? And if so, are some frequencies more informative than others? To investigate these questions, we develop a framework for quantifying the usefulness of correlations between odour signals at various frequencies, based on computing the Fisher information that they contain about intersource distance.

Our analysis of high bandwidth information requires consideration of the statistics and driving mechanisms of the range of signal frequencies present in naturalistic odour plumes. Odours released into turbulent flow environments are advected downstream with the mean flow. Concurrent distortion of odour streams by chaotic flow structures spanning a range of sizes (a defining characteristic of fluid turbulence) produces discrete odour filaments whose structure is subsequently altered by the turbulent mixing field. The associated stretching and folding of filaments sharpens odour concentration gradients, enhancing molecular diffusion in response to these flow-sharpened gradients and acts to broaden them [21]. The net effect of these competing processes, collectively constrained by odour mass conservation, is broadly referred to as chaotic advection, turbulent diffusion, or simply *mixing* [22, 23], and the resultant product is a complex, spatiotemporally dynamic concentration landscape. We refer to these as *odour landscapes* or *plumes*.

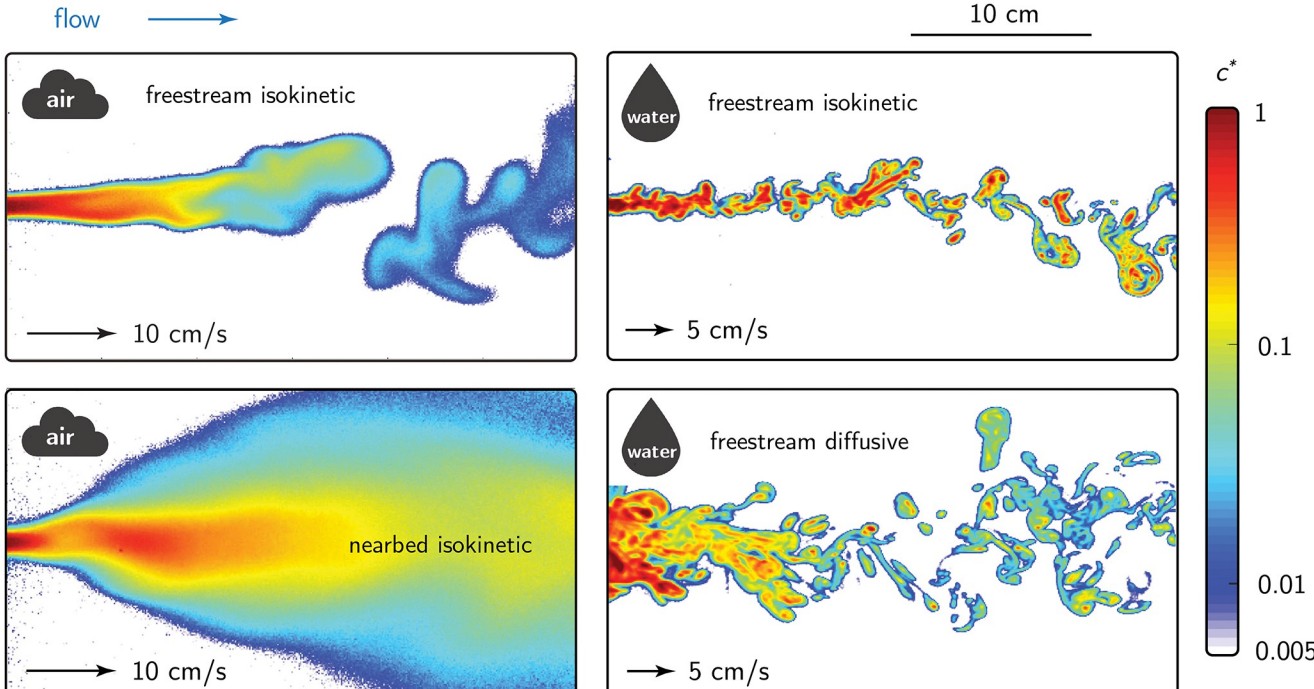

**Fig 1. Variety of odour landscapes.** Diverse single odour landscapes in air and water (adapted from [29]) with varying source release configurations and flow, fluid, and odourant properties. Animals may have to operate in multiple landscapes. For example, the odour landscape relevant to a mouse may be similar to the nearbed isokinetic condition (lower left) when it is searching the soil, and more like the freestream isokinetic (upper left) when it is rearing. The plumes were measured experimentally using planar laser-induced fluorescence (PLIF) and are described in detail in [30] (air) and [31] (water).

Statistics describing concentration fluctuations of passive scalar plumes (composed of odours that do not modify the local flow field) and their spatial variation are set by a number of dimensionless parameters. These include a Reynolds number (Re) describing the ambient flow environment (setting the statistics of the turbulent flow field) and a Péclet number (Pe) describing the relative importance of advective and diffusive odour transport (or alternatively a Schmidt number describing the relative diffusivities of fluid momentum and odour mass). Higher Re and Pe numbers are associated with more turbulent flows producing plume statistics dominated by turbulent diffusion. Additionally, a number of ratios describe the *source configuration*. These include its size relative to the largest, energy-containing eddies (characterized by an integral length scale), its proximity to solid boundaries, and its injection momentum relative to the flow. Source configuration effects are well-known (e.g. geometry [24, 25], source momentum relative to the mean flow [26], and proximity to surfaces [27, 28]). The variety of experimentally-measured odour landscapes shown in Fig 1 illustrates the effects of some of these important factors.

Plume statistics exhibit characteristic variations with distance from the source—including with streamwise and cross-stream location, parallel and normal to the mean flow vector, respectively [32]—owing to transitions between phenomenological regimes defined by differing sources of concentration variance. For example, for a small relative source size, near the source where the plume width is small relative to the large eddies, concentration variance is primarily driven by meandering of the whole plume driven by those large eddies (e.g. [33]). As the plume width grows with distance from the source, the large eddies produce less meandering and instead a range of eddy sizes are effective at mixing odour-laden and ambient (clean) fluid—a turbulent diffusive regime with enhanced relative dispersion of the plume locally

around its meandering centroid. A number of studies have examined statistical structure with distance from the source both experimentally [5, 34] and analytically [16]. Broadly speaking, source configuration effects influence near-field plume statistics (close to the source), whereas the far-field structure is largely self-similar, being set by the physics of turbulent flows and the linearly coupled advection-diffusion equations governing odourant transport and diffusion.

When multiple non-reactive odours are released from initially-distant locations into turbulent flows, odour-specific plumes evolve independently and uniquely under the influence of chaotic coherent flow structures [35]. The nature of the cumulative odour landscape describing all local odour concentrations and associated gradients is then simply the superposition of these odour-specific plumes, information that is potentially exploitable by a navigating sensor or organism. Of particular relevance to the work presented here is that, in addition to the factors setting the overall statistics of each individual odour plume, pointwise concentration correlations between odour species will depend strongly on the initial source separation distance [36, 37]. In particular, the correlations of concentration fluctuations can vary non-monotonically with distance from the source, and they show strong source separation effects with weak Reynolds number effects [38, 39]. This contributes to the observations of zero, negative, and positive correlation regimes (no interaction, destructive and constructive interference, respectively) [40, 41]. As an analogue to the single-source case where the local plume width relative to the large eddy sizes is important in driving local production of concentration variability, the source separation relative to the large eddy sizes in the multisource case drives these complex correlation behaviors.

Many statistical models describing concentration fluctuations for the single-source case have been proposed in the literature, and most leverage the simplicity of the exponential family to describe the one-point probability density functions [42, 43]. While the best match for a particular distance from the source and plume realization (Re, Pe, source configuration) varies, there is recent evidence that the flexibility of the Gamma distribution can robustly account for much of this variability [44], where the shape parameter is related to the concentration fluctuation intensity [24]. Intuitively, good statistical descriptors for concentration fluctuations must also capture the intermittent nature of concentration dynamics in turbulent flows manifesting as high probabilities of zero concentration events [43, 45]. Extensions of these models to describe the concentration statistics of multiple interacting sources are less frequent in the literature, but the exponential family was also shown to provide a good description of the distributions for the sum of concentrations [36, 46]. Even fewer studies have examined the spectral properties of correlations; however, recent work showed that for a single source separation, correlations increase with distance from the source and become more spectrally uniform [41].

Due to the dynamics described above, odour plumes in naturalistic environments contain fluctuations at a wide range of frequencies, modulated by distance to the source, proximity to solid boundaries, odour source characteristics, and the ambient flow environment. At the same time, recent research has demonstrated the high frequency acuity of olfactory systems, but the purpose of such sensitivity is unknown. One possibility is that information about inter-source distance is better encoded in different frequency bands depending on the olfactory context, and that the sensitivity of animal olfactory systems to rapidly fluctuating odours allows them to use this information when performing odour source localization. Thus the question we aim to answer in this study is: *What is the spectral distribution of the information that correlations carry about source separation?*

## 2 Results

To provide a testing ground for our approach to quantifying information in odour plumes we performed computational fluid dynamics (CFD) simulations of two-dimensional fluid flowing

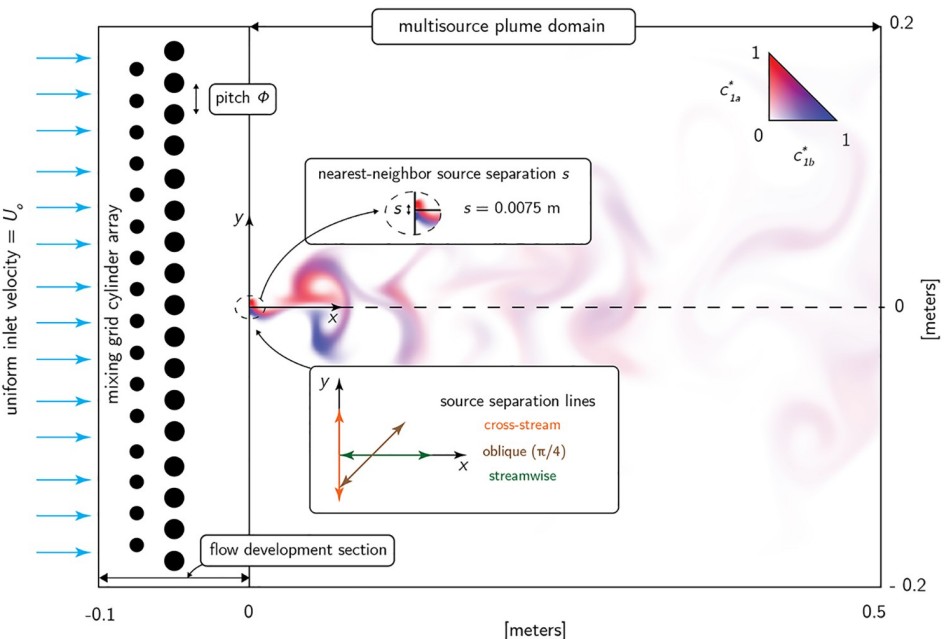

**Fig 2. CFD simulation domain.** The wind tunnel domain used to simulate multispecies odour landscapes. The total domain consists of an inlet flow development section followed by a plume domain section, beginning at $x = 0$. Most of the analyses in the Main Text are for an array of source pairs located at $x = 0$ and mirrored about the domain centerline at $y=0$ (the 'cross-stream' source configuration, orange arrows in the inset). We also performed analyses for sources placed oblique to the flow (brown arrow in the inset) or parallel to it ('streamwise', green arrow). Geometric, flow, fluid, and odourant properties are summarized in Table 1.

past equally spaced cylindrical obstacles in a rectangular wind tunnel, as shown in Fig 2. The interaction of the flow with the obstacles generated complex flow patterns that we used as a proxy for turbulent advection. We placed multiple odour sources in the simulated wind tunnel and measured the correlation of the resulting concentration profiles—henceforth referred to as *plumes*—at fixed probe locations downwind. A snapshot of the plumes generated by two adjacent odour sources is shown coloured blue and red in Fig 2.

Most of the results we present below are for odour sources placed 'cross-stream' i.e. at a fixed distance from the flow inlet, symmetric along the midline, and equally spaced transverse to mean flow. The direction along which the sources are placed is indicated by the orange arrow in the inset of Fig 2. The resulting odour signals that these sources generate are measured at a single probe location situated along the midline. To determine the generalizeability of our results we also performed our analyses for sources placed oblique to the mean flow (brown arrow in the inset) and parallel to the flow ('streamwise', green arrow), and with odour signals measured at 8 other probe locations. Further, we repeated these analyses for a supplementary set of simulations using similar flow parameters and source configurations but slightly different domain geometry. In Supporting Information Sec S3.1 in S1 File we describe these supplementary simulations and present some of the corresponding results of the analyses presented in the Main Text. The full set of source geometries and probe locations that we used are shown in S10 Fig.

Another snapshot of a flow pattern, this time for the maximally separated sources in the cross-stream configuration, is shown in Fig 3A. In Fig 3B we show some example concentration time series from sources at three different intersource distances. We express all distances in terms of the spacing between the obstacles, called the 'pitch,' and indicate this by suffixing

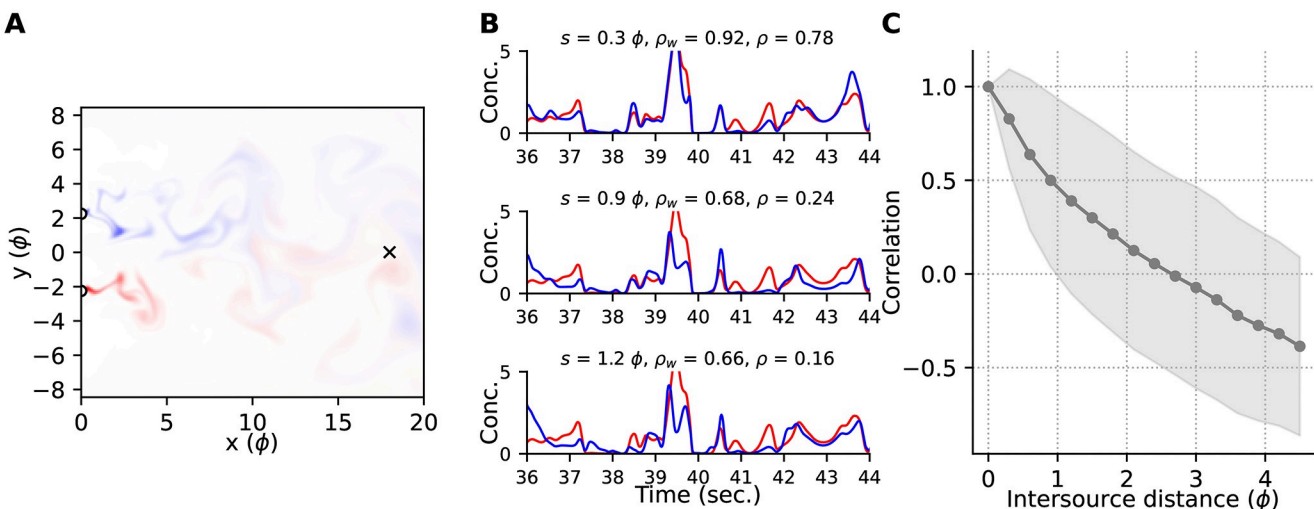

**Fig 3. Example plumes and correlations. (A)** Snapshot of the simulated flow patterns advecting two sources (colours). The plumes are sampled downwind at the probe location marked '×'. **(B)** Concentration profiles of the two odours at the probe location, scaled by the larger of their two standard deviations computed over the window shown. Sources were placed at the same downwind (horizontal) location and separated by cross-stream (vertical) distances indicated by $s$. Pearson correlations computed for the time windows shown, and for the entire simulation duration, are indicated by $\rho_w$ and $\rho$, respectively. **(C)** Mean ± standard deviation of the Pearson correlations computed over 1-second boxcar windows overlapping by 500 msec., for all sources at the intersource distances indicated. All distances are measured in pitch ($\phi$).

all distances with '$\phi$'. This is a natural length scale for distance normalization since it approximates the sizes of the largest eddies.

We used the Pearson correlation to measure the cofluctuations of two plumes. For the examples shown in Fig 3B correlations decrease with increasing intersource distance. We confirmed that this effect holds for the rest of our data in Fig 3C, similar to previous observations in the literature, e.g. [47].

The Pearson correlation of two plumes $x(t)$ and $y(t)$ in a time window of width $T$ seconds is

$$r = \frac{1}{T} \frac{\int_0^T (x(t) - \overline{x})(y(t) - \overline{y}) \, dt}{\sigma_x \sigma_y} \tag{1}$$

where $\overline{x}$ and $\sigma_x$ are the mean and standard deviation of $x(t)$ over this window, and similarly for $\overline{y}$ and $\sigma_y$. A key property of the Pearson correlation for our purposes is that it can be decomposed as a sum of correlations computed for each harmonic of the fundamental frequency $1/T$

$$r = \sum_{n=1}^{\infty} r_n, \tag{2}$$

where $r_n$ is the contribution from the $n$'th harmonic,

$$r_n \triangleq \frac{1}{2}(a_n c_n + b_n d_n), \tag{3}$$

and $(a_n, b_n)$ and $(c_n, d_n)$ are the cosine and sine coefficients of the Fourier decomposition of $x(t)$ and $y(t)$, respectively, scaled by the standard deviations of these two signals (see Eq 59 in Methods).

The spectral decomposition of plume correlations clearly depends on the corresponding decomposition of the plumes themselves. Observation of long duration signals such as plumes over short time windows introduces uncertainty about their spectral content called *spectral*

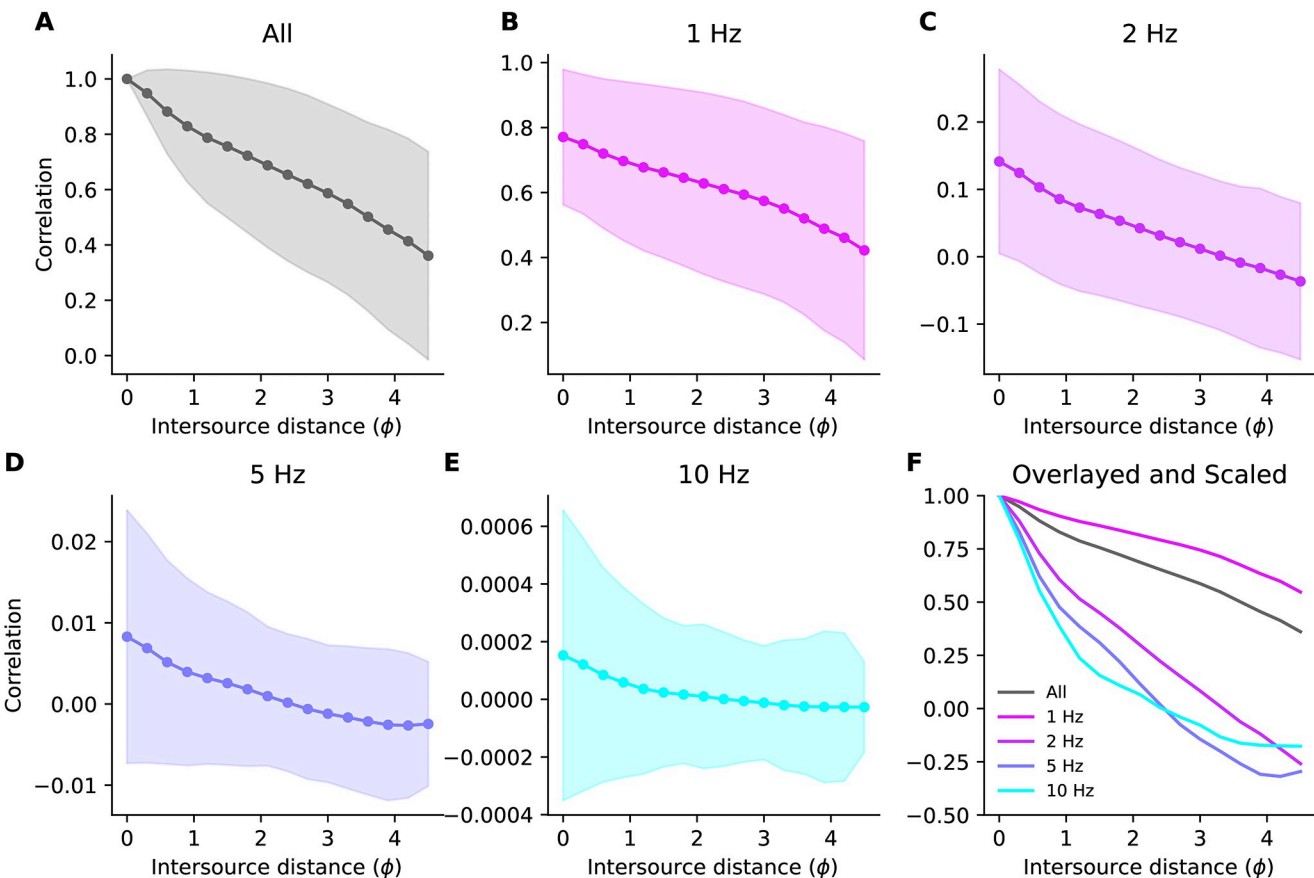

**Fig 4. Decomposition of correlations. (A)** Mean ± S.D. of Pearson correlations between the plumes emitted by two sources the specified distance apart, computed using 1-second Hann windows (compare to Fig 3C which used 1-second boxcar windows). **(B–E)** The components of the correlation at the frequencies stated above each panel. **(F)** The means of the data in panels (B–E) scaled to have the same value at intersource distance zero and overlayed for ease of comparison.

*leakage* [48], wherein the spectral content of the observed signal at a given frequency is not just that of the original signal at that frequency, but includes weighted contributions from all other frequencies. The effects of spectral leakage can be ameliorated by judicious weighting of the observations, called 'windowing'. Unless stated otherwise, the results below use a 1-second Hann window. In Fig 4 we have plotted the distribution of some of these correlation components as a function of distance.

To measure the amount of information that correlations $r_n$ provide about intersource distance $s$, we used the Fisher information [49], defined as

$$\mathcal{I}(s) = -\int \frac{\partial^2 \log p(r_n|s)}{\partial s^2} p(r_n|s) \; dr_n,$$ (4)

where $p(r_n|s)$ is the distribution of correlations $r_n$ for harmonic $n$ for an intersource separation $s$. Informally, the Fisher information measures the dependence of our estimate of $s$ on the observed correlations $r_n$. The more sensitive this estimate is to correlations, the more informative the correlations. The Fisher information also bounds the precision of unbiased estimation of intersource distance from correlations. Thus the higher the Fisher information, the higher the maximum precision with which intersource distance can be estimated from correlations.

Note also that the Fisher information is a function of the intersource distance $s$. Thus the information provided by correlations will in general depend on the (true) intersource distance.

## 2.1 Outline of our approach

Our approach to determining the spectral distribution of information in the correlations is as follows. First, we

1. Use simplifying assumptions about the statistics of the Fourier coefficients of the component waveforms to determine analytically tractable approximations to the distribution $p(r_n|s)$ needed to compute the Fisher information (Sec 2.2, Eq 21, and Fig 7);

    Although we make some of these assumptions for mathematical tractability rather than because we believe they hold in actual flows, we nevertheless test their validity numerically in Sec S4 in S1 File.
    Next, we

2. Use the distribution of correlations thus derived to compute analytic expressions for the Fisher information (Sec 2.3 and Eq 23);

3. Fit tractable distributions for the quantities required in the analytic expressions of Fisher information to our simulation data (Eq 24, Figs 8 and 9);

4. Compute the Fisher information using fitted parameters for each harmonic and intersource distance (Eq 26 and Fig 10).

## 2.2 The distribution of correlations

To determine an analytically tractable expression for $p(r_n|s)$, we began by using the rules of probability to relate the correlation $r_n$ to the Fourier coefficients $a_n$, $b_n$, $c_n$ and $d_n$. In our description so far we have assumed a single pair of sources at a given distance $s$ apart. In reality, the distribution of correlations is formed by combining the contribution of all pairs of sources separated by that distance. For example, in our simulations, there will frequently be many pairs of sources a given distance apart. To quantify effects that depend only on the relative location of sources, rather than the specific locations of specific sources, we made two location-independence assumptions.

First, we assumed that sources were close enough together relative to the animal and to the geometry of the flow so that

1. The distribution of coefficients from each source is the same for all sources, and

2. The distribution of coefficients from one source given those at another source depends only on the distance between them.

    We tested these assumptions numerically and found that they both broadly held (see S2 and S3 Figs).
    We show in Methods Sec 5.3.3 that these two assumptions allow us to write the distribution of correlations at a given source separation as

$$p(r_n|s) = \int p(r_n|a_n, b_n, c_n, d_n, s) \, p(a_n, b_n) \, p(c_n, d_n|a_n, b_n, s) \, da_n b_n c_n d_n. \qquad (5)$$

The first term in the integrand describes the dependence of correlations $r_n$ on the Fourier coefficients and the intersource distance. From Eq 3 we know that $r_n$ is determined entirely by

the Fourier coefficients. We can express this fact probabilistically by concentrating all of the probability density at the value in Eq 3 using the Dirac $\delta$ function,

$$p(r_n|a_n, b_n, c_n, d_n, s) = \delta\left(r_n - \frac{a_n c_n + b_n d_n}{2}\right).$$  (6)

The second term in the integrand of Eq 5 is the probability of observing the pair of coefficients $a_n$, $b_n$ from a source. To arrive at analytically tractable solutions, we made our next assumption:

3. Concentration profiles are Gaussian processes.

This assumption implies that the joint distribution of coefficients from each source, $p(a_n, b_n)$, is a two-dimensional Gaussian distribution. We emphasize that we made this assumption for analytic tractability; turbulent flows are well known to exhibit non-Gaussianity, for example in their velocity fields a [50] or the concentrations of advected substances [43]. Indeed, when we tested this assumption numerically we found that at most locations in the plume it only held for frequencies below 10 Hz (see S4 Fig). We consider some of the implications of this non-Gaussianity in the Discussion.

To specify the mean and covariance of our assumed Gaussian distribution, we made our next assumption,

4. Concentration profiles are temporally stationary.

This means that the statistical properties of plumes do not change with time. Temporal stationarity has the following important implications for the Fourier coefficients (see Sec S6.1 in S1 File for details). First, it implies that the marginal distribution of the sine and cosine coefficients must be the same, that is

$$p(a_n = v) = p(b_n = v).$$  (7)

Second, it implies that the coefficients at non-zero frequency have mean zero, that is

$$\langle a_{n \neq 0} \rangle = \langle b_{n \neq 0} \rangle = 0.$$  (8)

Third, it implies that the coefficients are uncorrelated, so

$$\langle a_n b_n \rangle = 0$$  (9)

The expectations in Eqs 8 and 9 are over time windows. We tested all three implications numerically and found that they broadly held (see S5 Fig).

By combining our Gaussian process assumption with our assumptions of location independence and temporal stationarity we can completely specify the second term in Eq 5:

$$p(a_n, b_n) = p(a_n)p(b_n) = \mathcal{N}(a_n; 0, \sigma_n^2)\mathcal{N}(b_n; 0, \sigma_n^2),$$  (10)

where $\sigma_n^2$ is the marginal variance of the coefficients for harmonic $n$.

The third term in Eq 5 is $p(c_n, d_n|a_n, b_n, s)$ and expresses the probability of observing the pair of coefficients $(c_n, d_n)$ from the second source given the observed coefficients $(a_n, b_n)$ from the first source located at an intersource distance of $s$. To specify it, we first represented the joint distribution of coefficients $p(a_n, b_n, c_n, d_n|s)$, as the graphical model in Fig 5A. The marginal independence of the sine and cosine coefficients $a_n$ and $b_n$ in Eq 10 is reflected in the network through the lack of connections between $a_n$ and $b_n$. Importantly, the network indicates that the coefficients $c_n$ and $d_n$ are conditionally independent given the observed values of $a_n$ and $b_n$. This is because the only routes by which $c_n$ and $d_n$ can influence each other are through

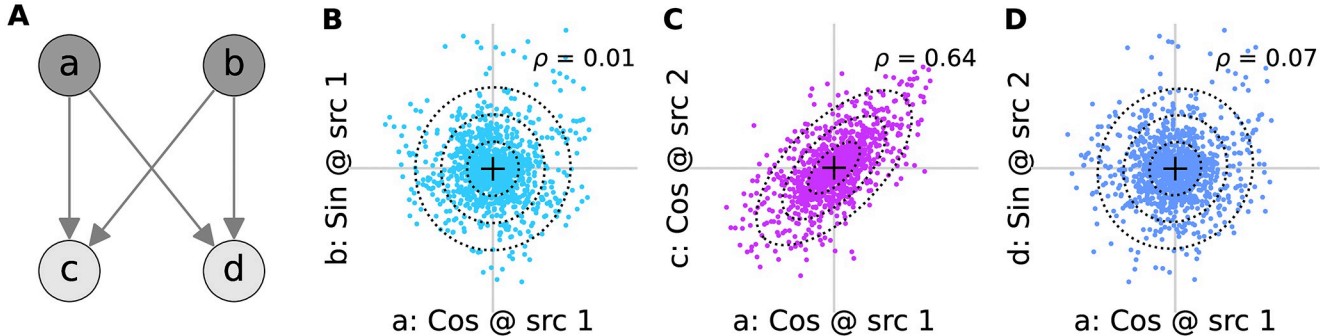

**Fig 5. Decomposition of phase relationships. (A)** Knowing the cosine ($a$) and sine ($b$) coefficients at one source can inform about the corresponding coefficients ($c$ and $d$, respectively) at another. The lack of arrows between $a$ and $b$ indicates their marginal independence. **(B)** Cosine ($a$) vs. sine ($b$) coefficients from the same source at 5 Hz, pooled across all sources. **(C)** Cosine coefficients from one source ($a$) vs. those ($c$) from the closest source in the positive vertical direction, pooled across all such pairs. **(D)** As in panel C but plotted against the sine coefficient from the neighbouring source. The Pearson correlation of the coefficients in each panel is indicated with $\rho$.

$a_n$ and $b_n$, which serve as common input to $a_n$ and $b_n$, and these are observed [51]. Therefore,

$$p(c_n, d_n | a_n, b_n, s) = p(c_n | a_n, b_n, s)\, p(d_n | a_n, b_n, s). \tag{11}$$

To specify a form for the conditional distributions on the right-hand side of Eq 2, we must determine the uncertainty that remains about the coefficients $c_n$ and $d_n$ of the component waveform at the second source when we know the corresponding coefficients $a_n$ and $b_n$ at the first. To do so, we expressed the component waveform at the second source as a scaled and phase-shifted version of the first, plus a residual. Doing so, we made our final assumption,

5. The conditional distribution $p(c_n, d_n | a_n, b_n, s)$ of coefficients at one source ($c_n$ and $d_n$) given those at another ($a_n$ and $b_n$), is bivariate Gaussian.

We performed a coarse test of this assumption and found that it held, at least within the limitations of this test; see S6 Fig.

With this final assumption in hand we concluded that the conditional distribution of the individual coefficients is

$$p(c_n | a_n, b_n, s) = \mathcal{N}(c_n; \beta_n(s)[a_n \cos\theta_n(s) + b_n \sin\theta_n(s)], \eta_n(s)^2), \tag{12a}$$

$$p(d_n | a_n, b_n, s) = \mathcal{N}(d_n; \beta_n(s)[b_n \cos\theta_n(s) - a_n \sin\theta_n(s)], \eta_n(s)^2). \tag{12b}$$

Here $\beta_n(s)$ and the $\theta_n(s)$ are the best-fit scaling and phase-shifts, and $\eta_n(s)^2$ is the variance of the residual (see Methods Sec 5.3.2).

Substituting these equations into Eq 11 and combining with Eq 10 we arrive at the form of the joint distribution of the coefficients

$$p(a_n, b_n, c_n, d_n | s) = \mathcal{N}([a_n, b_n, c_n, d_n]; \mathbf{0}, \mathbf{\Sigma}_n(s)) \quad (13a)$$

$$\mathbf{\Sigma}_n(s) = \sigma_n^2 \begin{pmatrix} 1 & 0 & \beta_n(s)\cos\theta_n(s) & -\beta_n(s)\sin\theta_n(s) \\ 0 & 1 & \beta_n(s)\sin\theta_n(s) & \beta_n(s)\cos\theta_n(s) \\ \beta_n(s)\cos\theta_n(s) & \beta_n(s)\sin\theta_n(s) & 1 & 0 \\ -\beta_n(s)\sin\theta_n(s) & \beta_n(s)\cos\theta_n(s) & 0 & 1 \end{pmatrix}. \quad (13b)$$

Note that by the location independence assumption, the marginal variance $\sigma_n^2$ of the coefficients does not depend on $s$.

The covariance in Eq 13 specifies two types of relationship between the coefficients from the two sources. Those that are 'in-phase' relate coefficients of the same type, for example the cosine coefficients $a_n$ and $c_n$. Those that are 'out-of-phase' or 'quadrature' relate coefficients of different types, for example the cosine coefficient $a_n$ at the first source and the sine coefficient $d_n$ at the second. Determining the spatial information that correlations provide requires incorporating both types of relationship by using the full set of covariances specified in Eq 13.

However, for our data the out-of-phase contribution is very small, and the in-phase relationship dominates. This can be seen, for example, in the high in-phase correlation of the data in Fig 5C vs. the near zero correlation of the out-of-phase data in Fig 5D (see also S17 Fig). Therefore, it will be convenient to focus on the marginal distribution relating only the in-phase coefficients. For the pair of cosine coefficients this is

$$p(a_n, c_n | s) = \mathcal{N}([a_n, c_n]^T; \mathbf{0}, \mathbf{\Sigma}_n^{\text{in}}(s)) \quad (14a)$$

$$\mathbf{\Sigma}_n^{\text{in}}(s) = \sigma_n^2 \begin{pmatrix} 1 & \rho_n(s) \\ \rho_n(s) & 1 \end{pmatrix}, \quad (14b)$$

where we've defined the in-phase correlation

$$\rho_n(s) \triangleq \beta_n(s)\cos\theta_n(s). \quad (15)$$

Note that this expression is not meant to constrain the dependence of $\rho_n$ on intersource distance $s$, and merely defines it in terms of the quantities $\beta_n$ and $\theta_n$ at that distance. The dependence on intersource distance can in principle be arbitrary, although we will find below (see Fig 8) that it is adequately described for small intersource distances by exponential decay.

The relationship relating the pair of sine coefficients $b_n$ and $d_n$ has the same form as Eq 14. Since the off-diagonal element of $\mathbf{\Sigma}_n^{\text{in}}(s)$ is the covariance of pairs of sine or cosine coefficients, we can use Eq 3 to relate the in-phase correlations to the average component correlations as

$$\langle r_n \rangle = \frac{\langle a_n c_n \rangle + \langle b_n d_n \rangle}{2} = \sigma_n^2 \rho_n. \quad (16)$$

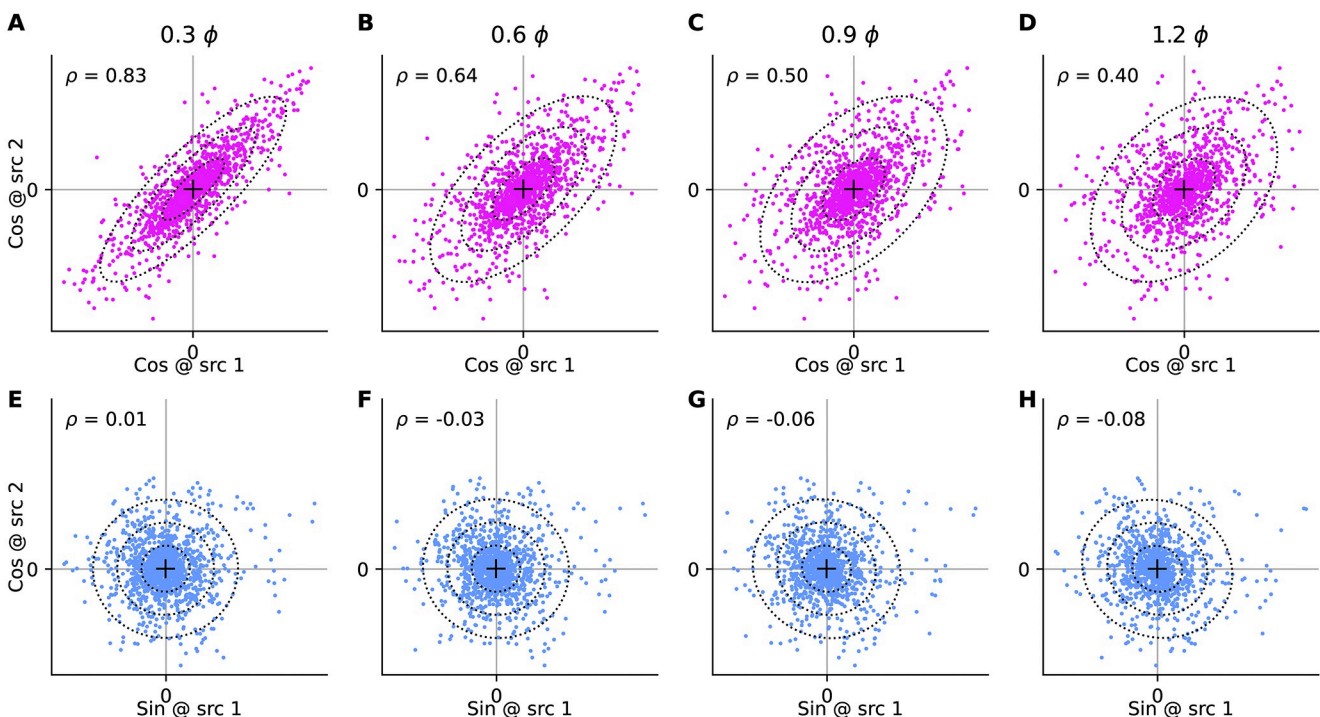

**Fig 6. Fourier coefficients vs. distance.** Distribution of the indicated coefficients for the 5 Hz component of the plume from one source vs. the indicated coefficient computed for the second source, computed for all pairs of sources separated by the intersource distance stated above panels A-D. Crosses indicate the mean of each dataset; ellipses indicate first three standard deviations of bivariate normal fit to the data; Pearson correlations ($\rho$) of the data in each plot are indicated in the top-left corner. **(A-D)** Cosine coefficient at source 1 vs. cosine coefficient at source 2. **(E-H)** Sine coefficient at source 1 vs. cosine coefficient at source 2.

The relationship of the out-of-phase coefficients can be described in a similar way. For example,

$$p(b_n, c_n | s) = \mathcal{N}([b_n, c_n]^T; \mathbf{0}, \boldsymbol{\Sigma}_n^{\text{out}}(s)) \tag{17a}$$

$$\boldsymbol{\Sigma}_n^{\text{out}}(s) = \sigma_n^2 \begin{bmatrix} 1 & \rho_n^{\perp}(s) \\ \rho_n^{\perp}(s) & 1 \end{bmatrix}, \tag{17b}$$

where the out-of-phase correlation is defined as

$$\rho_n^{\perp}(s) \triangleq \beta_n(s) \sin \theta_n(s). \tag{18}$$

The distribution $p(a_n, d_n | s)$ has the same form, but with negative the covariance between the two variables.

In Fig 6 we have plotted examples of the joint distribution of in-phase (panels A-D) and out-of-phase (panels E-H) coefficients. These plots confirm that the out-of-phase correlations, while not zero, are much smaller than the in-phase correlations.

Armed with the joint distribution of Fourier coefficients given intersource distance specified in Eq 13, we can return to Eq 5 and derive the distribution of correlations to be the

asymmetric Laplacian (see Methods Sec 5.3.3)

$$p(r_n|s) = \frac{1}{Z_n(s)} \begin{cases} e^{-2|r_n|/(Z_n(s)+\sigma^2 \rho_n(s))} & r_n \geq 0 \\ e^{-2|r_n|/(Z_n(s)-\sigma^2 \rho_n(s))} & r_n < 0 \end{cases}, \tag{19}$$

where the normalizing constant is defined as

$$Z_n(s) = \sigma_n^2 \sqrt{1 - \rho_n^{\perp}(s)}. \tag{20}$$

**A simplifying assumption.** Because the out-of-phase correlations of our data are typically very close to zero, in what follows we will assume that these correlations are zero. In that case, $Z_n(s) \approx \sigma_n^2$ and Eq 19 simplifies to

$$p(r_n|s) \approx \frac{1}{\sigma_n^2} \begin{cases} e^{-2|r_n|/\sigma^2(1+\rho_n(s))} & r_n \geq 0 \\ e^{-2|r_n|/\sigma^2(1-\rho_n(s))} & r_n < 0 \end{cases}. \tag{21}$$

To evaluate the agreement of the asymmetric Laplacian in Eq 21 with the observed distribution of correlations we compare their cumulative distribution functions (CDFs). In Fig 7A–7C we make the comparison for the 5 Hz correlation data at three different intersource distances. We quantified the agreement between the CDFs as 1 minus the largest absolute difference between them. A value of 1 would indicate perfect agreement, while, a value of 0, the smallest possible would indicate non-overlapping distributions. A heatmap of the match over the full range of intersource distances and frequencies (Fig 7G) reveals a match of $\sim 0.7$ and higher over most of this range.

Although there is good qualitative agreement between the data distributions of correlations and the corresponding asymmetric Laplacians of Eq 21, the plots in Fig 7A–7C also suggest that there are systematic differences between the two. In Fig 7D–7F we have plotted the differences between the data distribution and the asymmetric Laplacian fits. These plots have similar shapes, consisting of a narrow positive lobe followed by a broader negative lobe, eventually decaying to zero as both CDFs approach 1. This reveals that the data had more correlation values concentrated near zero, and correspondingly fewer large correlation values, than the asymmetric Laplacians.

To achieve a better fit ot the observed distribution of correlations, we elaborated the asymmetric Laplacian model in two ways to better capture small correlations (see Methods Sec 5.3.4). First, turbulent flows produce intermittent signals [50] so some very small correlations may be noise, not actual correlations. Therefore, similar to [43] for single-source concentration models, we incorporated intermittency using a binary random variable $z$ that indicated whether an observation $y_n$ was of a correlation $r_n$ drawn from our correlation model $p(r_n|s)$, or noise $w$:

$$y_n|z, r_n, w = zr_n + (1-z)w. \tag{22}$$

Second, we generalized the exponential decays in our correlation model of Eq 21 to the generalized inverse Gaussian $p(r) \propto r^{k-1} e^{-\alpha(r/\lambda+\lambda/r)/2}$ which also includes the Gamma distribution as a special case.

We fit both intermittent and non-intermittent versions of these models to the correlations at each frequency and intersource separation separately. Examples of such fits are shown in Fig 7, showing improved agreement with the observed correlations. The performance of the best-

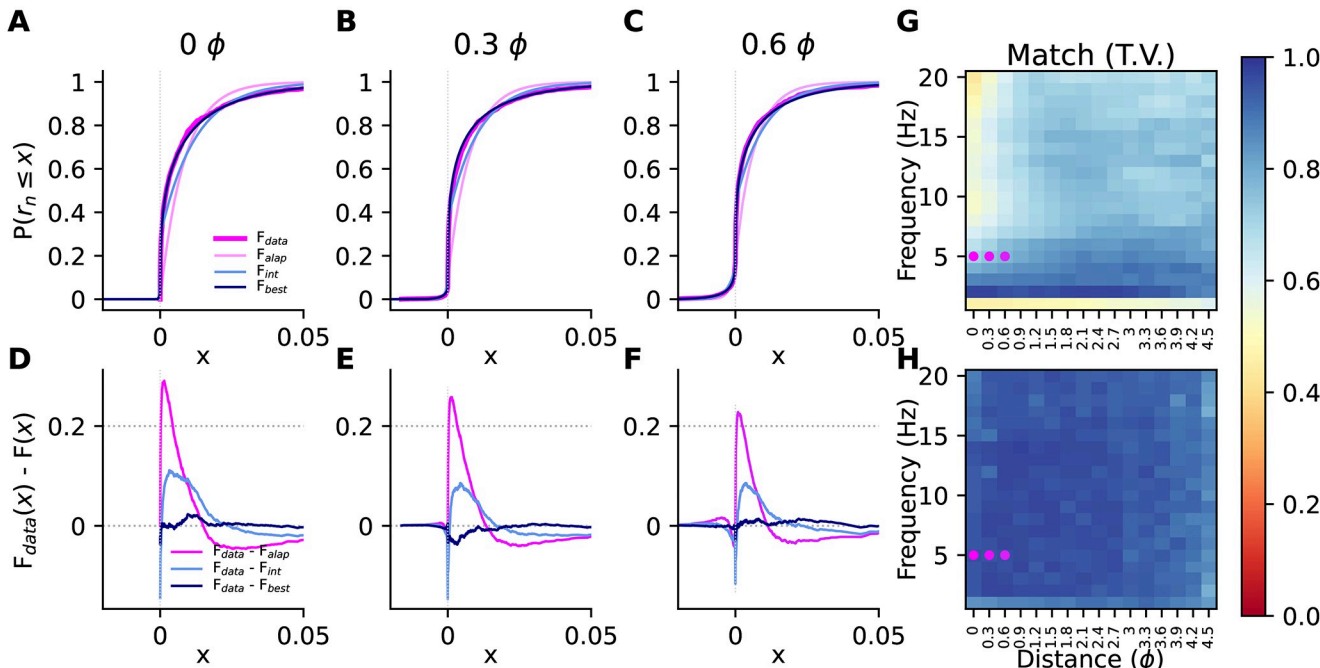

**Fig 7. Modeling the distribution of observed correlations.** (A–C) Examples of observed ($F_{data}$, thick pink) and predicted cumulative distribution function for the 5 Hz component of correlations, for three different intersource distances (indicated above each panel). Predictions were for the asymmetric Laplacian distributions ($F_{alap}$, pink) in Eq 21, asymmetric Laplacian with intermittency ($F_{int}$, light blue), and the best-fitting model when testing both intermittent and non-intermittent versions of the asymmetric Laplacian, Gamma, and generalized inverse Gaussian ($F_{best}$, navy). The parameters of $F_{alap}$ were derived from Gaussian fits to the distributions of Fourier coefficients, the parameters for the remaining fits were found by directly fitting the correlation data. (D–F) Absolute difference between the predicted and observed cumulative distribution functions in the corresponding panels in the first row. Data in A-F may be optimistic because fits were computed using some (though not all) of the data they are being qualitatively evaluated against in those panels. See panel H for performance on held-out data. (G) Fit quality for the asymmetric Laplacian model, measured as 1 minus the largest absolute difference between the predicted and observed CDFs, computed for all distances and frequencies listed. Higher values indicated better fits. Values may be optimistic because parameters were both trained and evaluated on the same (full) dataset. Coloured dots correspond to data points shown in the left three panels. (H) As in panel G, but for the best fitting model when testing both intermittent and non-intermittent versions of the asymmetric Laplacian, Gamma, and generalized inverse Gaussian, fitted directly to the correlations, and evaluated on unseen data. See Methods Sec 5.2.4.

fitting models on unseen data over the full range of frequencies an intersource distances are shown Fig 7H, demonstrating a very good fit over the entire range.

## 2.3 Computing the Fisher information

Having now determined expressions for the distribution of correlations given intersource distance, we can use Eq 4 to determine the Fisher information. Evaluating that expression requires not just a form for the correlation distributions themselves, but also for how they change with intersource distance. These changes in turn are determined by how the various parameters of our correlation model, Eq 19, change with intersource distance.

The three correlation models we have considered, the asymmetric Laplacian, Gamma and generalized inverse Gaussian, have 2, 4, and 6 parameters, respectively. The intermittent version of each model requires an additional two parameters to capture the intermittency. To avoid the complexity of modeling how large numbers of parameters change with intersource distance, we will use our simplest model, the non-intermittent asymmetric Laplacian of Eq 19, to estimate Fisher information.

The general expression using Eq 19 is complex but simplifies significantly when the out-of-phase correlation is zero, which is approximately the case for our data. Therefore in what

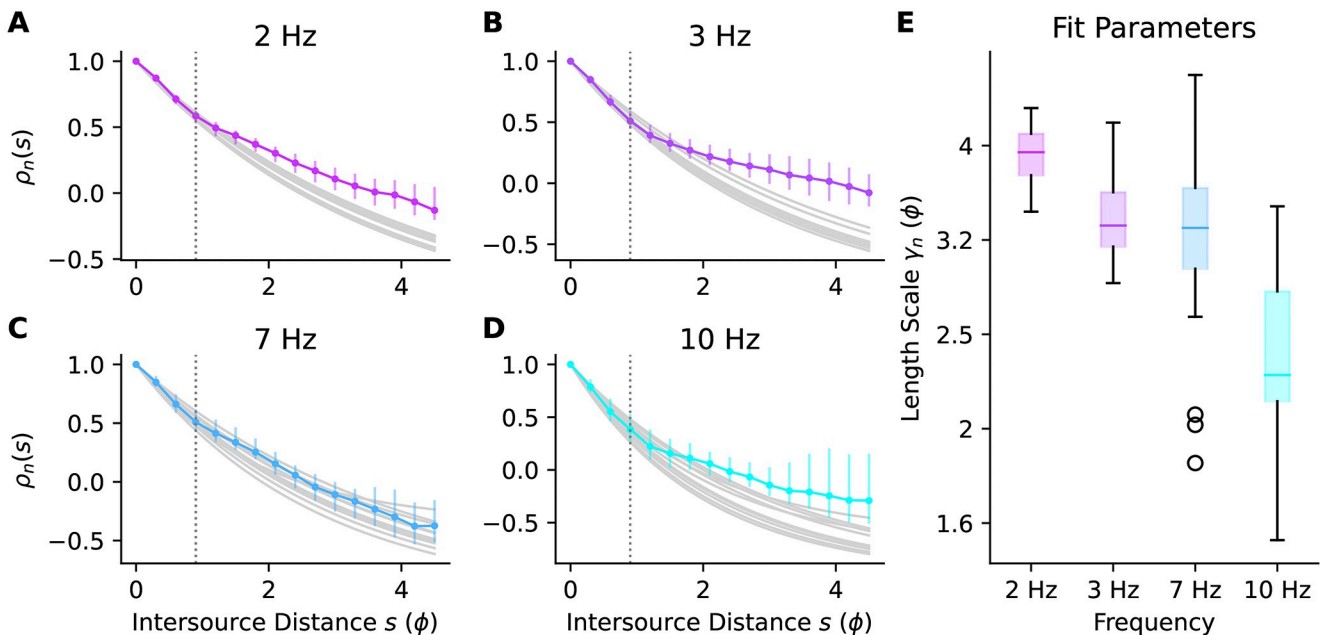

**Fig 8. Parametric fits to $\rho_n(s)$. (A–D)** Bootstrap medians (coloured points) and 5–95th percentiles (coloured bars) of $\rho_n(s)$ at each intersource distance, and a few example fits using bootstrap parameters (gray traces), for a few example frequencies. **(E)** Length scale parameter (on a logarithmic axis) for the fits to the data in (A–D). Boxes indicate inter-quartile range (IQR), central lines are medians, whiskers extend to last data points within 1.5 IQR, circles are outliers. Fits were computed for intersource distances up to 1 pitch, indicated by the dotted vertical lines in panels A-D.

follows, we will use the simplified expressions. The Fisher information is then (see Methods Sec 5.3.5)

$$\mathcal{I}(r_n, s) = \frac{2}{1 - \rho_n(s)^2} \left( \frac{d\rho_n(s)}{ds} \right)^2. \tag{23}$$

As basic checks of this expression we observe that

- It's non-negative since $0 \leq \rho_n(s)^2 \leq 1$;

- It depends on how the correlations change with distance, via the $d\rho_n(s)/ds$ term, so that when this dependence is zero, there is no information in the correlations, as expected;

- For a fixed amount of distance dependence $d\rho_n(s)/ds$, information is least when there is no correlation ($\rho_n(s) = 0$).

Armed with Eq 23, we still require the distance dependence of correlations $\rho_n(s)$ to compute the Fisher information. To motivate a parametric form for this dependence, we require that at the very least it should peak at an intersource distance of zero and decrease with distance. A simple parametric form that meets this requirement is exponential decay with distance $s$.

$$\rho_n(s) = (1 - b_n)e^{-s/\gamma_n} + b_n. \tag{24}$$

The two parameters of the model are the length scale of the decay $\gamma_n$, and the constant offset $b_n$ that determines the value at very large intersource distances. In Fig 8 we demonstrate the fit of this parametric form to some example correlation data.

Interestingly, when we examined the decay of correlation with distance, we observed that higher frequencies decayed faster than lower frequencies (Fig 9A). We plotted the length

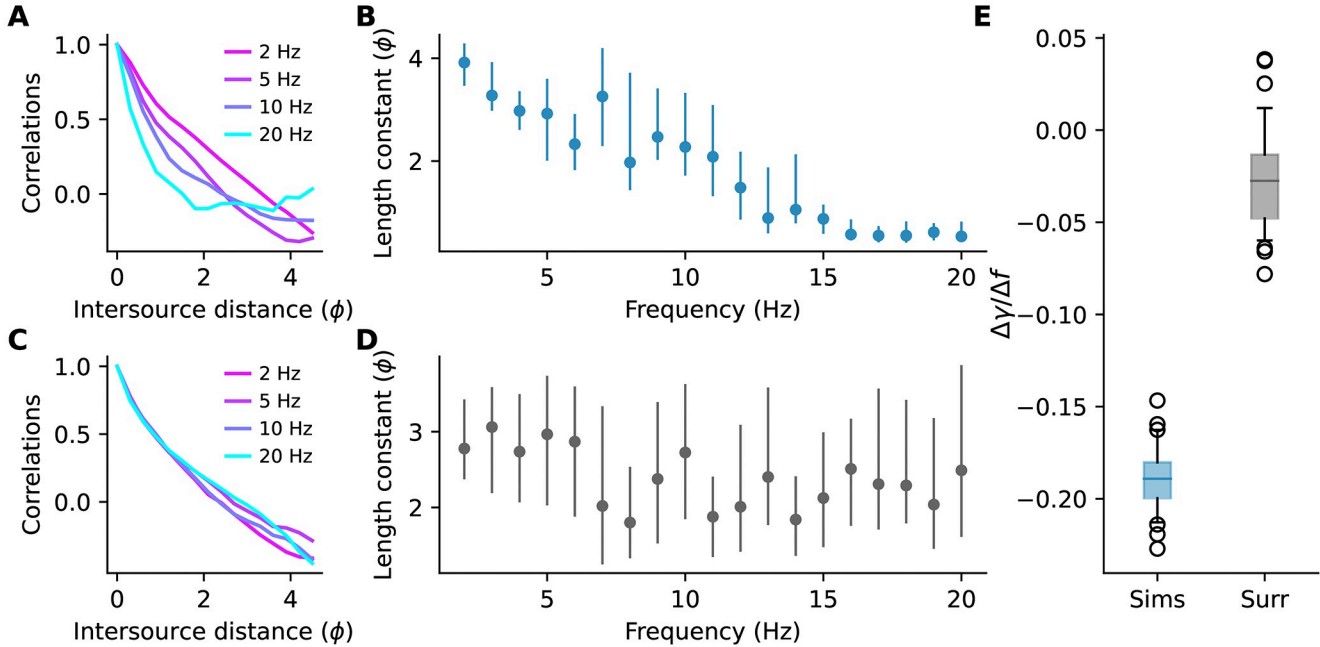

**Fig 9. The decay of correlations with frequency. (A)** Normalized correlations at different frequencies. **(B)** Length constants $\gamma$ of exponential fits to the decay of correlations for different frequency components. Dots and lines are the bootstrap median and 5th-95th percentiles, respectively **(C,D)** As in panels A and B but for surrogate data where all frequencies are equally informative. **(E)** Bootstrap distribution of coefficients when regressing the length constants in **(B,D)** on frequency. Negative values indicate length constants shorten with frequency. Central lines are medians, boxes are interquartile range, whiskers are [5th- 95th] percentiles, circles are outliers. The simulation data (but not the surrogate data) had very large time constant at 1 Hz so only the data from 2—20 Hz are shown above and used in the analysis.

constants of the generalized exponential fits to the decays against frequency (Fig 9B) and observed that they decreased at about one pitch every 10 Hz (Fig 9E). According to Eq 23, Fisher information depends on the rate of change of correlations with intersource distance. Therefore, the differences in decay rates that we observed suggest that the amount of information carried by correlations at different frequencies varies. As a comparison, we generated surrogate data with a similar power spectrum to our simulated plumes, but for which all frequencies were equally informative (see Methods Sec 5.2.7). We observed very little change in length constants with frequency in this surrogate data (Fig 9C–9E).

Given the parametric form Eq 24 of the distance dependence we derived an analytic expression for the Fisher information. To simplify the expression we first normalized distance $s$ by the length-scale parameter $\gamma_n$ and defined

$$s_n = \frac{s}{\gamma_n}. \tag{25}$$

In Methods Sec 5.3.5 we show that the Fisher information about intersource distance provided by the $n$'th harmonic component of correlations can be expressed in terms of normalized distance $s_n$ as

$$\mathcal{I}(r_n, s) = \frac{2}{\gamma_n^2} \frac{(1 - b_n)e^{-2s_n}}{(1 + b_n + (1 - b_n)e^{-s_n})(1 - e^{-s_n})}. \tag{26}$$

In Fig 10A we've plotted the bootstrap median and 5th-95th percentiles of Fisher information at a range of distances for a few example harmonics. The heatmap in Fig 10B shows the Fisher

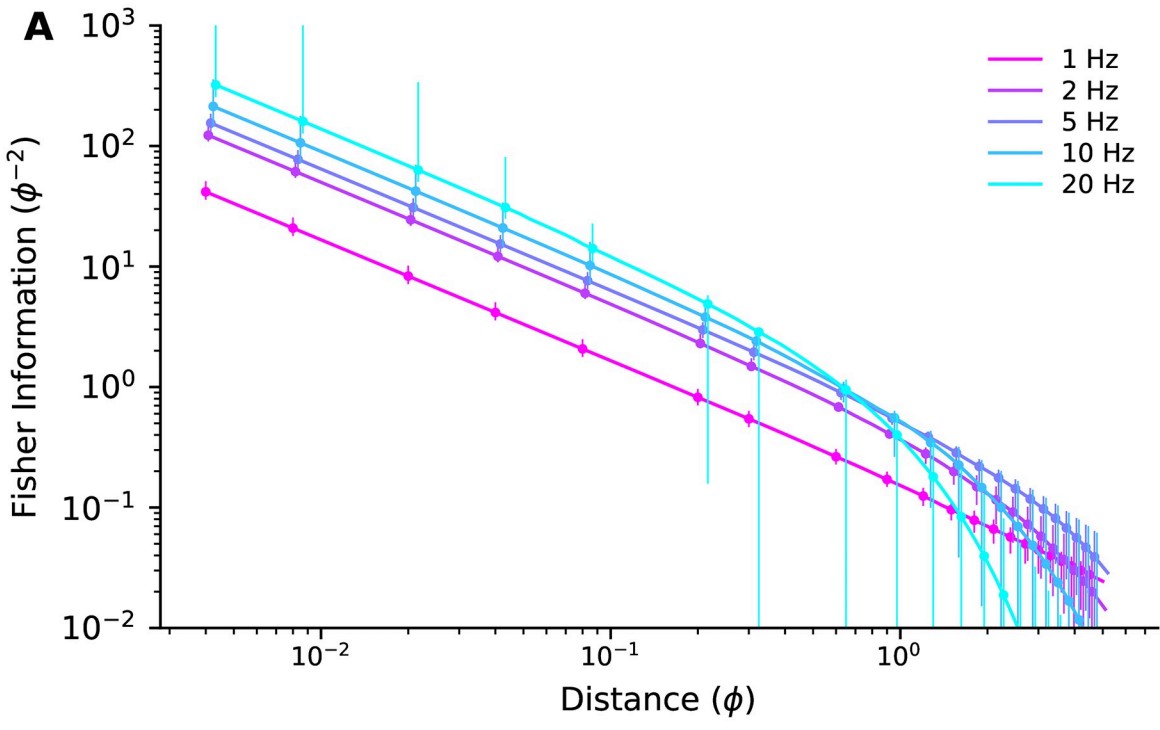

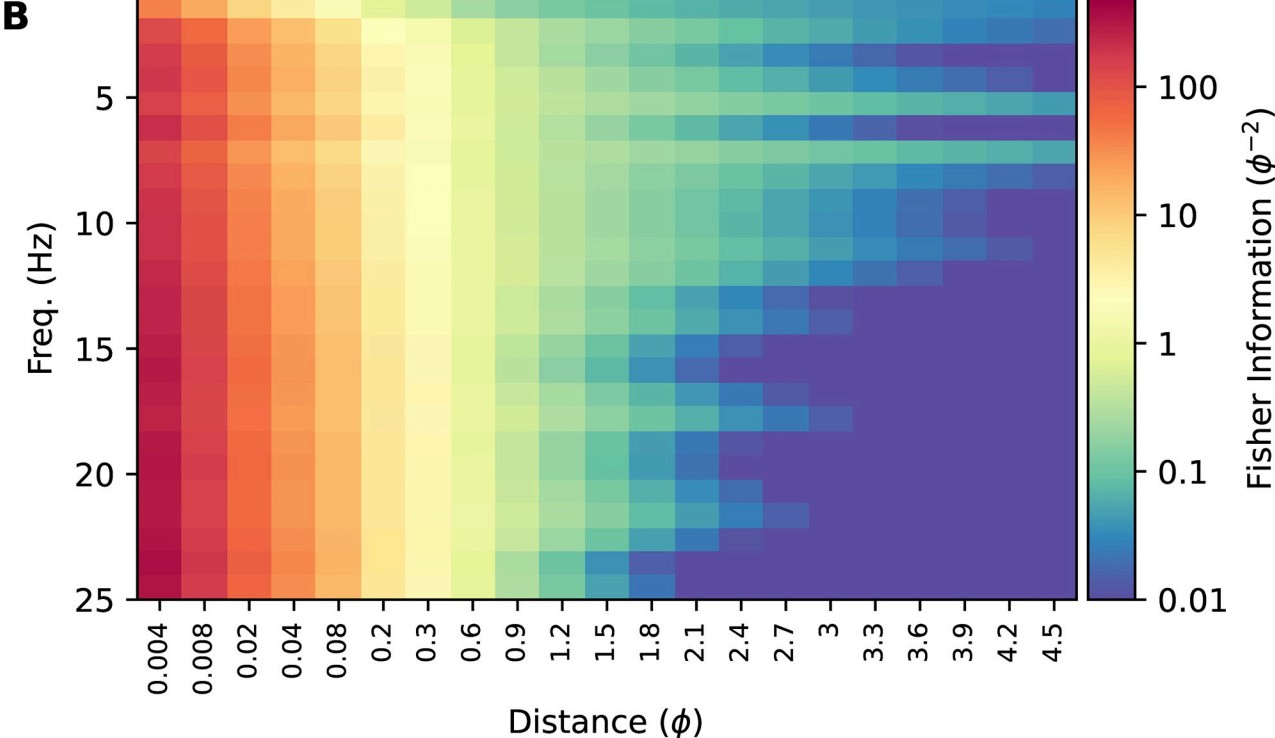

**Fig 10. Fisher information vs. intersource distance.** **(A)** Bootstrap medians (points) and 5–95th percentiles (bars) of Fisher information vs. intersource distance for five example frequencies. **(B)** Fisher information heatmap, computed using all the data (i.e. not bootstrapped). Note the nonlinear scaling of the x-axis.

information computed using all the data (i.e. not bootstrapped) at a broad range of frequencies and intersource distances.

**2.3.1 Which frequencies are more informative?.** The Fisher information heatmaps for our CFD plumes suggest that high frequencies are more informative when sources are close together. To see what such heatmaps would look like when the ground-truth distribution of information over frequencies is known, we generated surrogate plume datasets with similar power spectra to our CFD plumes, but with different distributions of information across frequencies (see Methods Sec 5.2.7).

To do so, we used the fact that the Fisher information depends on the in-phase correlations $\rho_n(s)$ according to Eq 23, and that these in turn are determined by the correlation (equivalently, covariance, since coefficients have zero mean) the in-phase Fourier coefficients by Eq 14b. We therefore generated surrogate data for $R$ odour sources at a given harmonic by sampling from a zero-mean $R$-dimensional Gaussian and adjusted the correlation of its dimensions, representing odour sources, to decay with intersource distance to produce the desired Fisher information at that harmonic.

In the first such dataset, all frequencies were set to be equally informative. To achieve this, we set the correlation between two surrogate sources to decay at the same exponential rate with their intersource distance at each harmonic (see Table 2). In Fig 11B we have plotted the Fisher information heatmap for this dataset. The information content varies with intersource distance, but is homogeneous with frequency, as expected, and unlike that of our CFD plumes (reproduced in Fig 11A).

Next, we generated a second surrogate dataset in which correlations decayed with intersource distance six times faster for the higher half of the frequency range than the lower half. This should result in the high frequencies being more informative at small intersource distances. At large intersource distances, high-frequency correlations will have decayed to zero due to their fast decay rate, yielding little information. Low frequencies will not have decayed completely to zero and will remain informative. Thus we expected to see high frequencies being more informative at small intersource distances, and low frequencies more informative at large intersource distances. This is indeed what we saw when we plotted the Fisher information heatmap for this dataset in Fig 11C.

The similarity of the information heatmap for our CFD plumes (Fig 11A) to that of the surrogate dataset where high-frequencies were *a priori* more informative (Fig 11C), and its dissimilarity to the heatmap of the dataset where all frequencies were *a priori* equally informative, supports the conclusion that for our CFD plumes high frequencies are more informative when sources are close together.

To emphasize the importance of windowing to these results, in Fig 11D we plot the Fisher information heatmap for our CFD plumes, but when analyzed with a 1-second boxcar window. The boxcar window, also known as a rectangular window, weights all samples equally. The poor spectral leakage properties of this window coupled with the power-law power spectrum of our data (S14 Fig) results in the low-frequency information masking that of the higher

**Table 2. Summary of surrogate datasets and the power and correlation functions used.** The kernel for each dataset was the product of the power and correlation functions: $k(i, j, n) = S(n)G(|i - j|, n)$.

| Surrogate Dataset | Power $S(n)$ | Correlation $G(|i - j|, n)$ |
|---|---|---|
| All harmonics equally informative,data power spectrum | $P(n, \alpha = 4)$ | $2\exp(-|i - j|/12) - 1$ |
| High frequencies more informative,data power spectrum | $P(n, \alpha = 4)$ | $\exp(-|i - j|/R(n))$ |
| High frequencies more informative,flat power spectrum | $P(n, \alpha = 0)$ | $\exp(-|i - j|/R(n))$ |

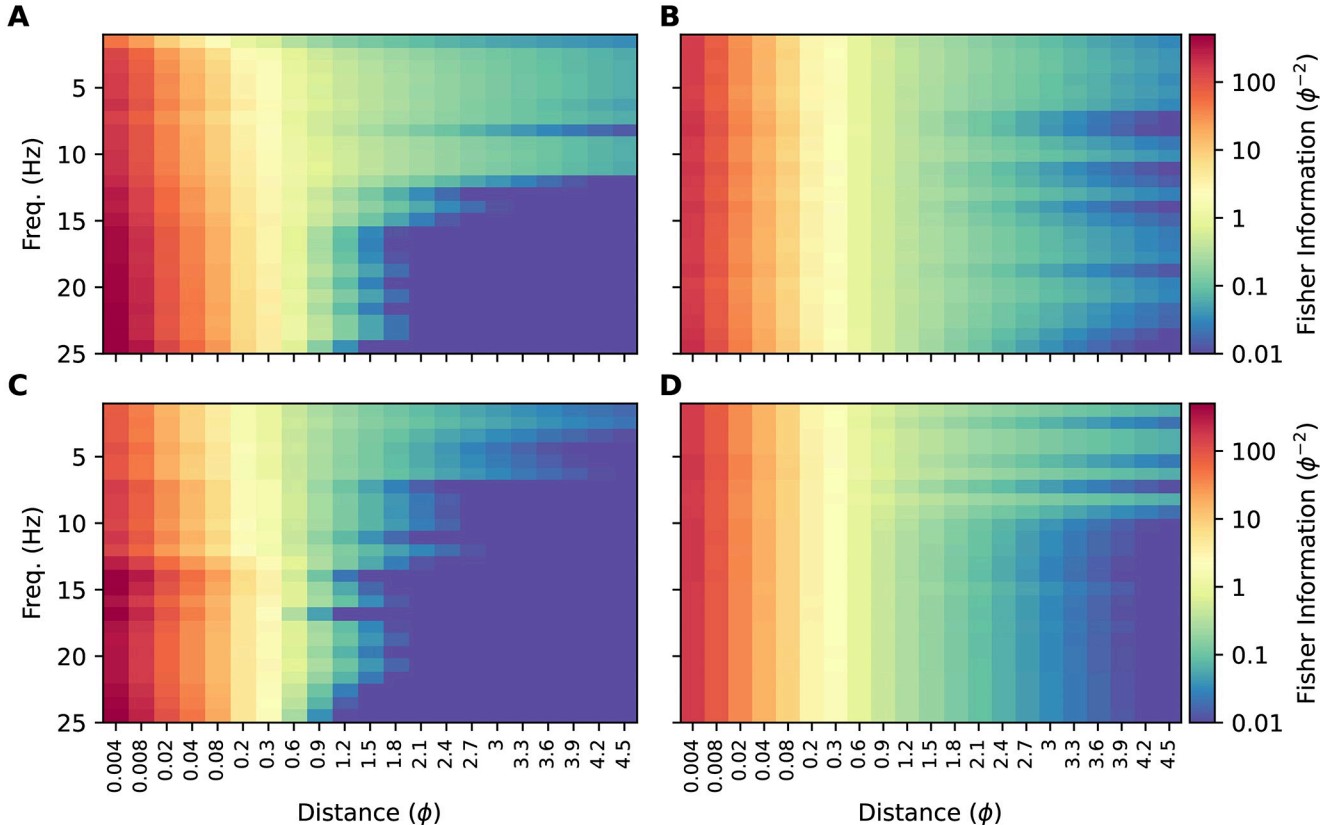

**Fig 11. Fisher information for simulations and surrogate data. (A)** Fisher information heatmap for our simulated plumes (reproduced from Fig 10B). **(B)** Heatmap for surrogate data with power spectrum similar to our simulated data and where all frequencies are equally informative. **(C)** Heatmap for surrogate data with power spectrum similar to our simulated data and where frequencies >12.5 Hz have the same, higher value of Fisher information than the lower frequencies. **(D)** Fisher information for our simulated plumes when using a 1-second boxcar window.

frequencies. This then results in an heatmap where information appears to be homogeneously distributed across frequencies, similar to Fig 11B.

From Eq 23, Fisher information depends on the rate at which correlations change with distance. The fact that correlations decay faster for high frequencies than for low frequencies means that when sources are closer together, the correlations at high frequencies will be more informative. This is shown in the left-most panel of Fig 12A, where we have plotted the information available at each frequency for an intersource distance of ∼0.1 pitch. At that intersource distance information shows a positive trend with frequency. As sources move farther apart, high-frequency correlations will have decayed, eventually changing at the same rate with distance as the slower decaying low-frequency correlations. This will result in all frequencies being similarly informative, as shown in the middle panel of Fig 12A for an intersource separation of ∼1 pitch. As sources move even farther apart, the lower frequencies will become more informative, as shown in the right panel of Fig 12A for an intersource separation of ∼2 pitches. The overall change in the trend as intersource distance increases is shown in Fig 12C and shows a distinctive elbow shape, so in what follows we will frequently refer to such data as 'elbow plots.' For surrogate data in which all frequencies are equally informative, trends in information with frequency are minimal at all intersource separations; see Fig 12B and 12C. Note that the exponential decay in correlations with frequency means that regardless of which frequencies are more informative, the absolute amount of information available will decrease

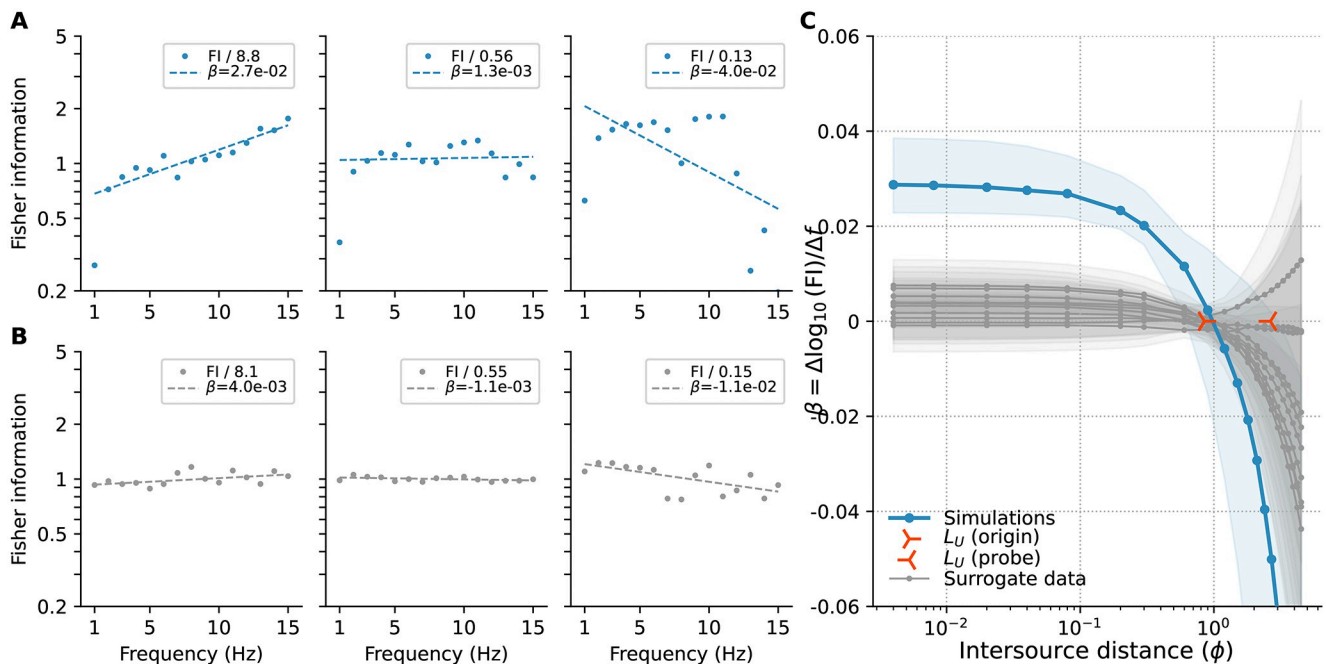

**Fig 12. Fisher information vs. frequency. (A)** Base 10 logarithm of the Fisher information at each frequency component (dots) and linear fits (dashed lines), for intersource separations of $\sim 0.1$, 1, and 2 pitches (left, middle and right panel, respectively), for the simulated plumes using a 1-second Hann window. Coefficients $\beta$ of the linear fits are stated in the legends. Data have been scaled by the amount shown in each legend to ease comparison. **(B)** As in panel A but for surrogate data where all frequencies are equally informative. **(C)** Bootstrap median (dots) and 5th-95th percentiles (bands) of the coefficients $\beta$ of linear fits to the logarithm of Fisher information regressed on frequency for the CFD plumes (blue) and 10 surrogate datasets where all frequencies are equally informative (gray). Positive coefficients indicate information increases with frequency, negative coefficients that it decreases. Centres of orange markers indicate the integral length scale ($L_U$, see Methods Sec 5.2.8) in the vertical direction (orthogonal to the mean flow) of the vertical velocity field, computed at the midline where the odour sources were located ('origin', $L_U = 0.85\phi$) and at the probe location ('probe', $L_U = 2.6\phi$). Linear fits were computed using robust regression, see Methods Sec 5.2.5.

with intersource distance. This is because information is derived from the rate of change of correlation with intersource distance, and this decreases as sources move farther apart. This effect is shown by the decrease in the down-scaling applied to the data in the panels of Fig 12A and 12B.

The analyses above relied on the use of windows that reduced spectral leakage when the amount of power in different frequency bands is different, as is the case in real plumes and in our simulations. By using the Hann window to reduce leakage we were able to distinguish the different rates at which correlations decay in different frequency bands, and therefore the different amounts of information that they contain. As we show in S7 and S8 Figs these effects were not limited to the 1-second Hann window used above, and we also observed them when we used other windows that similarly reduced spectral leakage.

**2.3.2 Testing other probe locations and source geometries.** The results we have presented above were computed at a single probe location, located on the midline of the flow downstream of the odour sources. Because animals will find themselves oriented at a variety of locations relative to odour sources of interest, it is important to determine the extent to which our results generalize to other locations within the plume. Therefore, we selected 8 additional probe locations within the plume, chosen to be dispersed enough to test the generalizability of our results, while also staying within the plume and avoiding boundary effects (see S1 Fig). These locations are indicated with coloured '×'s in Fig 13A. The blue '×' marks the probe

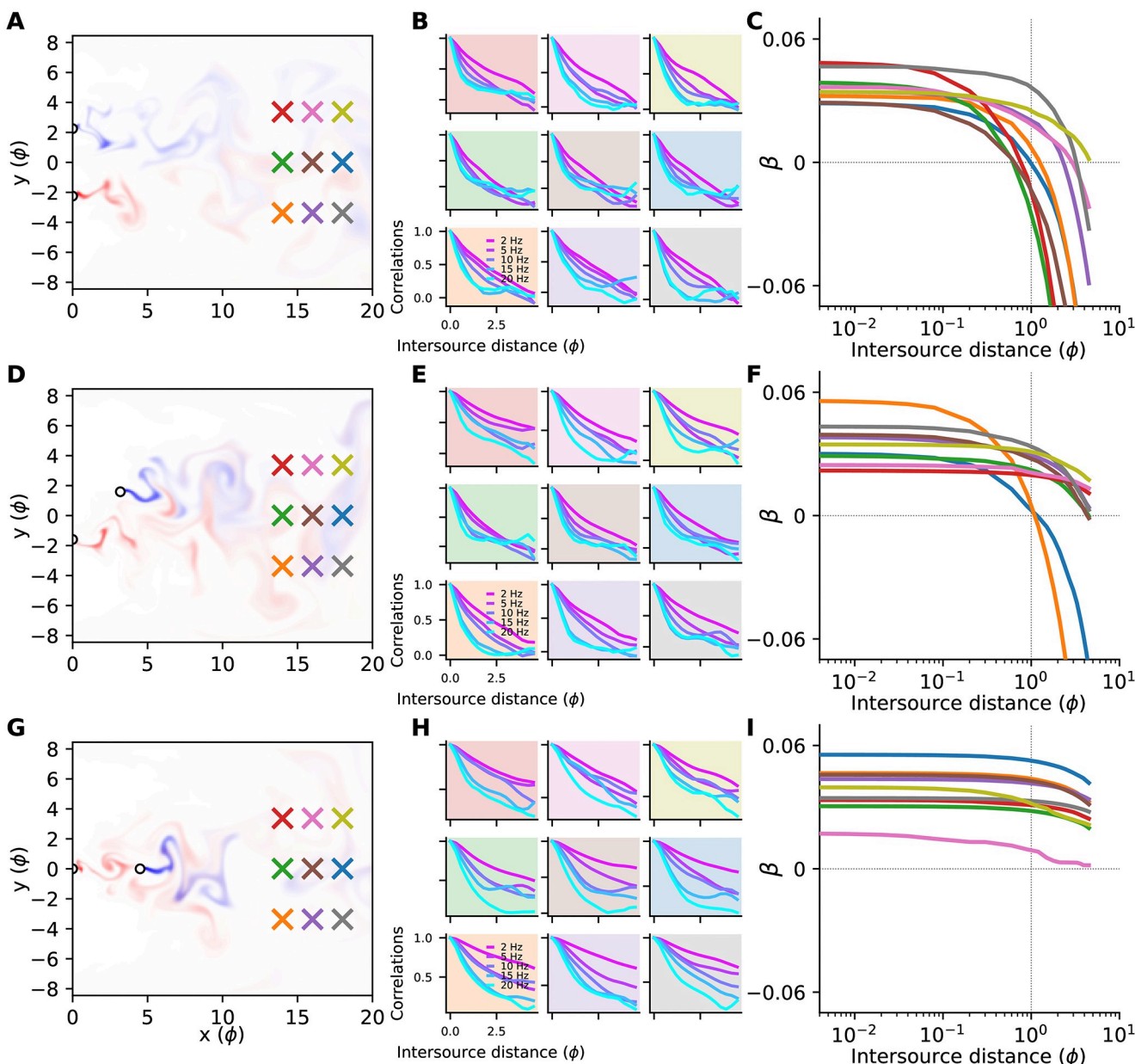

**Fig 13. Results for other probe locations and source geometries.** (**A**) Plume snapshot and probe locations for the cross-stream source geometry. Sources are equally spaced at *x* = 0 along a line transverse to the mean flow. The plumes (colours) are generated by the most distal pair of sources (white 'o's). Probe locations are marked with coloured '×'s. The cross-stream source geometry is the one analyzed in most of the Main Text, using the data measured at the blue probe location. (**B**) The decay of correlations with intersource distance for a few example frequency bands, at the nine different probe locations in panel A, indicated by the background colour of each panel. The data in the blue panel is for the principal probe location analyzed in the Main Text and was previously shown in Fig 9A. (**C**) Elbow plots for each probe location in panel A, and coloured accordingly. The blue curve is for the principal probe location analyzed in the Main Text and is the same as in Fig 12C. Only the bootstrap medians, and not the confidence intervals, are shown, for clarity. (D,E,F) As in panels A-C, but for the oblique source configuration, where sources are equally spaced along a line at 45-degrees to the mean flow. (F,I) As in panels A-C, but for the streamwise source configuration, where sources are equally spaced along a line parallel to the mean flow.

location for the results we've presented so far, while the remaining '×'s mark the new probe locations within the plume.

We performed the analysis describe in the previous section for each new probe location, culminating in 'elbow' plots like those of Fig 12C indicating whether high or low frequencies

were more informative at each intersource distance. The panels in Fig 13B show how correlations decay with intersource distance at each probe location. These panels reveal that at every probe location high-frequency correlations decay faster than lower frequency correlations when sources are close together. In Fig 13C we have overlayed the resulting elbow plots for all probe locations, showing only the medians for clarity, and coloured according to the probe locations in Fig 13A. The blue curve is the mean data previously shown in Fig 12C, while the remaining curves show the results for the new probe locations. All of the curves show the same qualitative behaviour: each starts at a positive value for $\beta$ at low intersource distance, indicating that high frequencies are more informative than low frequencies when sources are close together. As intersource distance increases, all the curves trend downwards, and most eventually reach negative values of $\beta$, indicating that low frequencies become more informative when sources are sufficiently far apart.

Our results thus far are for sources located perfectly 'cross-stream', transverse to the mean flow. Natural odour sources will assume many other orientations relative to the mean flow. It is therefore important to determine whether our results generalize to other source orientations. To do so, we performed simulations using two new source orientations.

In the first, odour sources were equally spaced on a line at 45 degrees to the mean flow. In Fig 13D we have plotted a snapshot of the plumes generated by two most distant sources in this configuration. The panels in Fig 13E show how correlations decay at each probe location for this source configuration, again revealing that high frequencies decay faster than low frequencies when sources are close together. The resulting elbow plots of Fig 13F reflect this and show that at all probe locations in this source configuration, high frequences are more informative than low frequencies when sources are close together.

In the second new source configuration, we placed our sources parallel to the flow (streamwise). Fig 13G shows the plumes generated by the most distant sources in this configuration. The correlation decay results in Fig 13H reveal the same qualitative trends as in the corresponding panels for the previous two configurations, with high-frequency correlations decaying faster than those at low frequencies when sources are close together. This is reflected in the the elbow plots of Fig 13I which show that high frequencies are more informative than low frequencies when intersource distances are low.

In summary, our analysis of 8 new probe locations and two new source configurations all reveal the same same qualitative trends that we observed in our original probe location and source configuration, namely that high-frequency correlations decay faster than low-frequency correlations when sources are close together (compare Figs 9A, 13B, 13E and 13H), resulting in high frequencies being the most informative when sources are closer together, with the balance shifting towards low frequencies as intersource distance increases (compare Figs 12C, 13C, 13F and 13I). We saw the same qualitative trends when we performed our analyses using longer windows (2-second Hann, S21 Fig), shorter windows (0.5-second Hann, S22 Fig) and windows with different shape (1-second Kaiser-16, S23 Fig).

Finally, we repeated our entire procedure for a supplementary set of simulations, and observed the same qualitative effects (S24 Fig), though there were exceptions for some probe locations and source configurations. For example, high frequencies decay more slowly than low frequencies for small intersource distances at the red and brown probe locations in the oblique plume (S24E Fig), which may contribute to low frequencies being more informative for nearby sources than high frequencies at the red probe location (note the inverted elbow in S24F Fig). Another prominent example comes from the same source configuration but analyzed using 500 msec. window, for which the elbows from a few additional probe locations are inverted (S26F Fig). However, all of these exceptions are for probe locations that are at the edge of the plume. This is partially due to the fact that odour sources are also closer to the

probes in the supplementary simulations. Both of these properties may contribute to the behaviour we observe at those locations.

When computing the Fisher information for the new source geometries above, we continued to use the in-phase component of the correlations. Initially, we had expected that the out-of-phase component of the correlations would play a larger role in these new geometries. For example, in the setting sources are arranged parallel to the flow direction, the velocity of the flow and the spacing between any pair of sources imply an initial phase difference between them, which may survive until the flow reaches the probe locations and manifest as strong out-of-phase correlations. However, we found that the magnitude of the out-of-phase correlations for the two new source configurations (S18D and S19D Figs) was small and similar to that for the original, transverse source configuration (S17D Fig), justifying our approximation of the Fisher information to use only the in-phase correlations for those source geometries as well.

## 3 Discussion

### 3.1 Quantifying information in plume correlations

The concentration timeseries from two odour sources measured at a downwind location tend to become more correlated when the sources move closer together. Therefore, correlations contain information about the spatial separations of the odour sources that generated them. Correlations can be decomposed as the sum of component correlations at different frequencies. Recently it was shown that mice can discriminate correlated vs. anti-correlated concentration timeseries at up to 40 Hz [20]. Could this high bandwidth sensitivity be caused by additional spatial information those frequencies might carry? Motivated by this question we set out to quantify the information contained in the component correlations about the spatial location of odour sources, to determine whether high frequencies contain more spatial information than lower frequencies.

Our approach is based on parametric modeling of how component correlations decay with intersource distance. We first express the component correlation at each frequency in terms of the corresponding Fourier decomposition coefficients of the odour timeseries being correlated. We then fit parametric models to these correlations and how they change with intersource distance. Using these parametric models we then derived a closed form expression for the Fisher information contained in the component correlations about the intersource distance in terms of the correlation of the Fourier coefficients and how these change with intersource distance (Eq 23).

We applied our approach to two computational fluid dynamics simulations of two-dimensional grid turbulence to determine which frequency bands were most informative about relative source locations. To verify our approach we also applied it to several surrogate datasets which we constructed to contain different patterns of spatial information in their component correlations. Analyzing our data using a 1-second Hann window to reduce spectral leakage, we observed that high-frequency correlations decayed faster than those at low frequencies (Fig 9). This meant that high frequencies were more informative when sources were less than $\sim 1$ pitch apart, and low frequencies when sources were farther apart than this Fig 12. We saw this effect for other similar leakage-reducing windows (S7 and S8 Figs), and not in surrogate data in which all frequencies were equally informative.

We tested the generalizability of our analysis by testing two additional source configurations and eight additional probe locations. In all of these cases, we found broad agreement with our orginal simulation, showing high frequencies were more informative than low frequencies Fig 13 when sources were close together. We observed the same qualitative effects when we tried different window sizes (S21 and S22 Figs) and window shapes (S23 Fig).

We repeated this procedure for our Supplementary simulations, and again saw the same qualitative effects as in the Main simulations for most source configurations and probe locations (see S24 Fig). There were, however, exceptions. For example, the high-frequency correlations at the red and brown probe locations in the oblique plume shown S24E Fig decay more slowly than the low-frequency correlations, which may contribute to high frequencies being less informative for discriminating sources close together using the plume signal at the red location. However, these and the other exceptions we found (e.g. in S26E Fig) all occured at the edges of the plume, where odour signals are more sparse. In addition, the sources in the Supplementary simulations are closer to the probes than those in our Main simulations. These two effects may account for the unusual signal properties we measured there.

In summary, in all 9 probe locations for all three different source geometries, three different window sizes and two window shapes, we saw the same effect in our Main simulations, namely, that high frequencies were more informative than low frequencies when sources were close together. We also observed this at most corresponding points in our Supplementary simulations, with the exceptions occuring at plume edges. We therefore believe our results are robust for probe locations within the plume.

**3.1.1 Implications of statistical assumptions.**   To determine the Fisher information contained in correlations about source separation, we required a probabilistic model of these correlations and how they vary with source separation. In our initial approach, we made simplifying assumptions about the Gaussianity of the plumes and their interactions and derived the predicted correlation distributions to be an asymmetric Laplacian (Eq 19). While this distribution fit the data well (Fig 7), we found that it deviated from the observations systematically due to its inability to describe the high probabilities of low correlation events (i.e. intermittency). This is perhaps unsurprising given that instantaneous plume distributions are known to be non-Gaussian [43] with the exception of large distances from the source, exhibiting transitions from exponential-like behaviors near the source through a log-normal transition to the far-field Gaussian regime [24]. To address these deficiencies in the correlation models, we then modeled the correlations directly using the asymmetric Laplacian as a template. We firstly extended it to incorporate intermittency by attributing very small correlations to noise. Secondly, in addition to the asymmetrical Laplacian, we also evaluated the ability of the Gamma and generalized inverse Gaussian distributions to describe the distribution of observations flagged as true correlations. We found that the extended models were able to capture the observed correlation distributions quite well across all source separations and frequencies (Fig 7H). The improved fits also indicate that the joint distribution of coefficients is not mulitvariate Gaussian. An interesting avenue for future investigation would be to determine the joint distribution of Fourier coefficients that would predict the same parameteric form for the distribution of correlations as our model when we fitted the correlations directly.

Ideally, we would have used these more sophisticated correlation models to compute the Fisher information; however, this computation would require describing how additional parameters depend on intersource distance, greatly increasing the difficulty of our analysis. We therefore used the simpler Gaussian assumption to compute the Fisher information. Would our findings hold if we used more accurate assumptions to derive Fisher information? Our simplified analysis suggested that an important quantity that determines Fisher information is the speed with which correlations decrease with intersource distance. A robust observation in our data, independent of any statistical assumptions, was that high-frequency correlations decreased more rapidly than low-frequency correlations. Because Fisher information is a measure of how rapidly the distribution of an observed quantity (like correlations) changes with respect to variations in estimated parameter (like intersource distance), the faster changes in high-frequency correlations with intersource distance that we observe will likely

mean that more sophisticated computations of Fisher information using more accurate statistical models will still yield our core observation that high frequencies are more informative than low frequencies at small source separations.

**3.1.2 Other considerations.**   We aimed to determine whether high frequencies contained more spatial information than low frequencies. We found that this was the case when sources were close together. At intermediate source separations, all frequencies were equally informative. Low frequencies were more informative when odour sources were far apart. However, the absolute amount of information in these latter cases was much lower, and presumably more easily obscured by noise. Thus, when correlations contain enough spatial information to rise above the noise floor, it may be contained mainly in high frequencies. Therefore the sensitivity to high-frequency correlations recently observed in mice [20] may endow animals with fine spatial resolution when locating odour sources in the environment. We note, however, that diffusion acts to smear out high-frequency information more quickly than low-frequency information, given the sharper concentration gradients associated with these fluctuations. The distance that high-frequency information can persist in a given olfactory context is therefore set by the mean wind speed, the strength of turbulent straining, and the odourant molecular diffusivity.

To test the generalizability of our results we additionally analyzed plumes generated by sources arranged oblique and streamwise to the mean flow. An interesting result of this analysis was that the roll-off of the elbows shifted to larger intersource distances as the source separations transitioned from cross-stream to streamwise configurations (compare Fig 13C, 13F and 13I). In our developing grid turbulence flows, local integral length scales grow with streamwise distance (see S1A Fig), and these length scales relative to the source size set the plume dispersion characteristics. Therefore, sources separated in the cross-stream direction are introduced into similar length scales in the flow and thus disperse similarly, while sources separated in the streamwise direction disperse under the influence of different length scales in the flow. Furthermore, for the array of fixed probe locations considered here, sources separated in the cross-stream, oblique, and streamwise directions are located at different relative streamwise and lateral distances from the probe locations. These two effects likely drive the different roll-off behaviors observed in the elbow plots for different source configurations. Future work could investigate these connections.

We observed that correlations dropped approximately exponentially with intersource distance, see e.g. Fig 3C. This exponential decrease with intersource distance has been previously observed in the fluid dynamics literature [52]. These correlations are composed of the component correlations at each frequency component. Therefore, there is considerable latitude in the way the component correlations may decay: they are bounded by the variance at each frequency and must sum at each intersource separation to the overall correlation at that separation. Nevertheless, we observed that the component correlations all had the same, nearly exponential form (Fig 9A). Explaining the reason why the component correlations decay exponentially would be an interesting topic of future work.

Our work highlights the importance of window shape when analyzing signals where information is distributed across the frequency spectrum and where spectral power spans a wide range. Our information measures were computed for 1-second windows. Longer windows would reduce leakage effects, but would lengthen the time needed for behvioural decisions. An interesting question is how the distribution of spectral changes depends on the time windows used—or in ethological terms, which frequencies the olfactory system should attend to if odour source localizations have to be made quickly. We leave the investigation of these important questions to future work.

Fisher information is a 'local' measure about how a given signal can distinguish a given parameter value from neigbouring values. Our expression for Fisher information Eq 23 is accordingly local and depends on the value of component correlations $\rho_n(s)$ at a given intersource separation and how they change around that point. To actually evaluate the Fisher information we used parametric fits to these component correlations (Eq 24). In contrast to the local nature of Fisher information, these parameteric fits are 'global' in that the fitted value at each intersource separation is influenced by all the data, not just those local to the separation. For example, deviations at large intersource separations can affect the length constant $\gamma$ of the fit, which will in turn affect the value of Fisher information reported for small intersource distances (and all others). It will therefore be useful to investigate whether fitting approaches that are more local, such as splines, produce better estimates of the Fisher information.

We chose to compare the two plumes by computing their Pearson correlations. We chose the Pearson correlation because it is insensitive to the mean and scale of the signals being compared and only registers their covariation, and it is this covariation that reflects the relative separation of the odour sources. By our choice of Pearson correlation we do not mean to imply that it is the optimal measure for decoding intersource distance. It is, rather, a simple measure whose information content provides a lower bound for the total information present in the interacting plumes about source separation. Other measures, for example those based on event timings (see e.g. [53]) may be more effective for odour source localization.

We used Fisher information as our information measure. An obvious alternative is mutual information between the intersource distances and correlations. One advantage of Fisher information is that it is based on the likelihood function $p(r_n|s)$ only and does not require the specification of a prior distribution on intersource distances. Mutual information requires the joint distribution $p(r_n, s)$ of intersource distances and correlations and thus *does* require a prior on intersource distances—or alternatively a prior $p(r_n)$ and likelihood $p(s|r_n)$ for correlations, which seem harder to specify. However, this is not a major shortcoming as a prior on intersource distances would not be hard to motivate or to determine empirically for a given environment. The type of information provided by Fisher and mutual information are also different. Fisher information is 'local' in the unknown parameter (intersource distance) and indicates how discriminable a given value of that parameter is from another value infinitesimally close. Hence it is a function of the unknown parameter, as seen in Fig 10. It also bounds the variance of unbiased estimates of intersource distance from correlations, and would thus be particularly relevant if the animal needs to accurately localize the relative locations of odour sources. Mutual information is a 'global' measure in that its computations involves averaging over the full joint distribution of intersource distances and correlations. We leave the task of determining which of these two measures of information is most relevant for quantifying the spatial information in plumes to future work.

## 3.2 Dispersion, coalescence, and coherent flow structures in naturalistic plumes

Our multisource plume datasets were generated in two-dimensional, spatially-decaying grid turbulence. The statistics of these plumes are broadly consistent with those observed in diverse naturalistic olfactory contexts, regardless of interesting phenomenological differences between two-dimensional and three-dimensional turbulent flows [54]. These consistencies include the observation of positive correlation regimes far from the source with clear source separation effects (Fig 3), the importance of intermittency in describing correlation distributions (Fig 7), the success of exponential-like distributions in describing correlations (Eq 19 and Fig 7), and

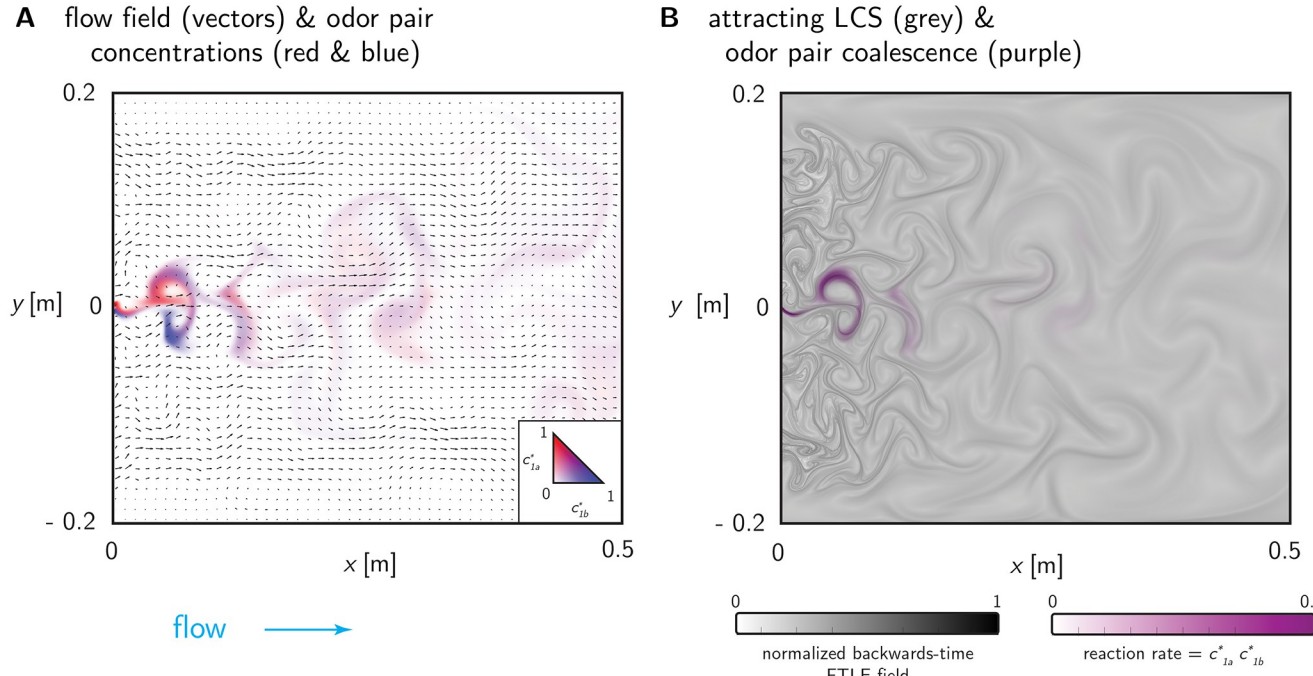

**Fig 14. Lagrangian coherent structures. (A)** An instantaneous snapshot of the flow field (vectors) and the concentration fields of a pair of odours ($c_{1a}$ in red and $c_{1b}$ in blue). **(B)** The corresponding attracting LCS quantified through the backwards-time FTLE field (grey) and "reaction rate" of the odour pair (the product of their concentrations, purple).

the exponential decay of correlations with increasing source separation (Fig 8). Given these broad consistencies, the analytical toolbox we present for quantifying the Fisher information in the spectral components of correlations about source separations is likely well-suited for applications to many odor landscapes relevant to diverse olfactory contexts.

Given the evidence for spectral information contained in correlations about source separation, what are the dynamical processes that drive these correlations? Lagrangian coherent structures (LCS) provide an intuitive framework for understanding the physical processes that drive coalescence of initially-distant odour sources and thus the statistics of their correlations. LCS arise from a dynamical systems view of fluid turbulence, where stable and unstable manifolds arise in regions of exponential fluid straining [55]. These manifolds represent repelling and attracting regions of the flow, respectively. Attracting structures are regions of disproportionate importance in odour dispersion because they closely correspond to spatial regions in which single odours evolve and initially distant odours coalesce in chaotic flow environments [56]. A representative instantaneous attracting LCS computed using the backwards finite-time Lyapunov exponent (FTLE) is shown in Fig 14 alongside the underlying flow-field. Ridge lines in the FTLE field show strong spatial correlations with odour pair coalescence.

The exact structures of LCS and their evolution in space and time will vary among turbulent flows occurring in diverse odor landscapes. The LCS dynamics will largely determine the correlations between initially-separate odour sources, and therefore the specific flow signature inherent in the LCS field will manifest in any analysis of plume information. Thus, we suggest that LCS be used to properly contextualize our analyses and findings in order to better relate our findings to naturalistic plumes. A full quantitative linking between spectral plume information and statistical characteristics of the LCS field could be an intriguing subject for future work.

### 3.3 Connections to optimal navigation and control

Our framework for quantifying spectral information about source separation may have implications for optimization of multi-source search strategies, both in an olfactory context and more widely in optimal navigation and control applications. In olfactory search, an important goal is not just to find *a* strategy to reach the target source, but to find the *optimal* strategy to locate the source, namely, one that minimizes time to arrival. Model-based approaches for search optimization rely on the agent possessing an internal model of probability distributions characterizing the environment (presumably through instinct in biological contexts). Concretely, recent research situated olfactory search within optimality theory by framing it as a partially-observable Markov decision process (POMDP), using Bayesian updates to source location belief based on an internal model of odor detection likelihood across plume regions [57]. This POMDP strategy optimized for a simplified single-source scenario, and in the more complicated multi-source case, integrating a model of inter-source distance likelihood based on high bandwidth signal correlations could prove useful in optimizing the search.

Even in model-free reinforcement learning approaches, the insights from our findings can be incorporated to enhance the agent's policy development. By structuring rewards around the acquisition of high-resolution odor signals and their correlations, agents can learn to incorporate information gain about inter-source distances in their strategies. The selection between model-based and model-free approaches involves consideration of the trade-off between strategies that are most efficient in specific environments, versus those that are more robust to changes in, or imperfect knowledge of, the search environment. Typically, model-based approaches tend to be more efficient if the search landscape matches the agent's internal model (i.e. result in lower mean arrival times), while model-free approaches tend to be more robust to external perturbations of the expected environment (i.e. result in shorter tails in arrival time distribution across a variety of scenarios) [58]. Our study could potentially improve optimal strategy development in either context, by improving the agent's internal model in model-based approaches, or by guiding reward structure and/or observation resolution in model-free approaches.

Optimal navigation is often situated in optimal *control* contexts, in which control policies guide an agent's movement based on continuous feedback from the environment [59]. Control policies could potentially benefit from leveraging high-bandwidth odor correlations to adjust the control policies dynamically. For instance, when high-frequency information indicates that sources are close together, control systems could fine-tune the agent's trajectory to exploit this data for rapid convergence to the source. Conversely, when sources are farther apart, control policies could be adjusted to rely more on low-frequency signals, which are more stable over longer distances. More generally, recent investigations of optimal control in the navigation of complex environments suggested that search strategies improve with the amount of information available to the agent [60]. Our findings identify an additional source of information about relative source locations available at the local level, information that may provide significant value in multi-source search control strategies.

## 4 Conclusion

In conclusion, we have outlined an approach for quantifying the spectral information in odour plume correlations about odour source locations. Our analysis revealed that high frequencies are more informative about intersource distance than low frequencies for sources that are close together. We tested the robustness of our findings at nine probe locations for three different source configurations, three time window lengths and two window shapes, using two different simulated two-dimensional turbulent flows, and saw the same qualitative effects in all

settings except occasionally at plume edges. Owing both to the variety of odour landscape characteristics in naturalistic plume environments and to the preeminent role of CFS in driving both dispersion and coalescence processes in natural as well as simulated plumes, we expect that the method and findings outlined here will also be applicable to describing the spatial information contained in three-dimensional plumes.

# 5 Methods

All of our simulation data, and the code for our simulationes and analyses are provided at https://github.com/stootoon/fisher-plumes.

## 5.1 Computational fluid dynamics

We numerically modeled two-dimensional grid-turbulence in a wide wind tunnel and introduced an array of unique odour sources to generate chaotic multisource plumes (Fig 2). These data allowed us to i. investigate spectral correlations between odour sources as a function of source separation, ii. compute spectral Fisher information content encoding source separation and location, and iii. evaluate spatial structure in these entities. To test the robustness of the analyses to generalized plume environments, we conducted a second set of numerical simulations using different flow and odour parameters yielding an additional plume dataset with different spatiotemporal characteristics. Below we detail the CFD methods used for all analyses in the Main Text. We provide an alternative set of supplemental data derived from other numerical implementations in the Supporting Information Sec S3.1 in S1 File.

**5.1.1 Model domain, simulations, and meshing.**   The model domain (Fig 2) consists of a wide rectangular wind tunnel (0.62 m x 0.4225 m, $x$ by $y$) with a mixing grid cylinder array distributed across the flow development inlet section (0.12 m in streamwise length). An array of 16 odour sources (8 source pairs mirrored about the domain centerline at $y=0$) of width $L_f$ was distributed spatially across a transect a short distance downstream of the mixing grid at $x = 0$, covering a range of source separations $s$ for analysis (Table 1). Source $y$-locations were constrained to the central 1/4 of the total inlet width to minimize walls effects, i.e. any impacts on cross-stream plume spread due to growth of the velocity boundary layer along the lateral walls. A uniform inlet velocity upstream of the mixing grid produces a chaotic flow field downstream through the nonlinear interactions of wake structures shed by the mixing grid. Simulations spanned a dimensional duration 70 s consisting of i. a 10 s start up period to establish fully chaotic flow conditions throughout the model domain and ii a subsequent 60 s analysis period containing dynamic multispecies plumes generated by the source array issuing into the chaotic flow environment.

Numerical simulations were performed via finite element discretization of the Navier-Stokes and continuity equations (Eqs S1 and S2 in S1 File) governing fluid flow, and the coupled, non-reactive advection-diffusion equation (Eq S4 in S1 File) governing odour transport and diffusion (see Sec S2 in S1 File). The COMSOL Multiphysics package (ver. 6.0) was used to generate the mesh in the model domain described below and to solve the system of equations resulting from the weak-form discretization of the governing equations, subjected to prescribed initial and boundary conditions described below. Geometric, flow, fluid, and odourant properties for model runs in the domain depicted in Fig 2 are summarized in Table 1.

The unstructured finite-element mesh contained 289,607 total triangular and quadrilateral elements (110 elements/cm$^2$ on average), locally refined in spatial regions with strong fluid velocity or odour concentration gradients, notably along the solid surfaces of the mixing grid cylinders and lateral walls and near the finite-width odour sources where strong initial odour gradients persist. The level of mesh refinement (increasing model degrees of freedom) was

**Table 1. Summary of model geometry, flow, fluid, and odourant properties used for multisource plume simulations.** Properties are for air and common airborne odourants at $\approx 20°C$.

| Model Geometry | |
|---|---|
| total domain size [m, $x$ x $y$] | 0.62 x 0.4225 |
| inlet section length, plume domain length [m] | 0.12, 0.5 |
| pitch $\phi$ [m] | 0.025 |
| small cylinder diameter $\phi_1$ [m] | 0.01 |
| large cylinder diameter $\phi_2$ [m] | 0.015 |
| **Flow, Fluid, and Odourant Properties** | |
| mean ambient flow speed $U_o$ [m/s] | 0.1 |
| fluid density $\rho$ [kg/m$^3$] | 1.2 |
| dynamic viscosity $\mu$ [Pa*s] | 1.80E-05 |
| odour molecular diffusivity $D$ [m$^2$/s] | 1.50E-05 |
| odour source concentration [mol/m$^3$] | 1 |
| odour source width $L_f$ [m] | 0.005 |
| nearest-neighbor source separation [m, on-center] | 0.0075 |
| centerline source $y$-locations at $x = 0$ [+/- m: start, spacing, end] | 0.00375, 0.0075, 0.05625 |
| **Key Dimensionless Parameters** | |
| Reynolds numbers Re [based on $\phi_1$, $\phi_2$, and $\phi$, respectively] | 70, 100, 170 |
| Schmidt number Sc | 1 |

iterated to a final spatial resolution sufficient to resolve the smallest anticipated fluid velocity and odour concentration gradients. The final mesh implemented for all model runs had typical maximum and minimum element sizes (spatial resolution) of approximately 2 mm and 0.2 mm, respectively, with approximately 35 and 10 mesh elements per pitch $\phi$ and per source diameter $L_f$, respectively.

**5.1.2 Initial and boundary conditions.** Initial conditions were $\mathbf{u}^* = 0$, $p^* = 0$ and $c^* = 0$ everywhere. The mixing grid and lateral walls were no-slip conditions, $\mathbf{u}^* = 0$, with zero total odour flux normal to the surfaces, $-\mathbf{n} \cdot (\mathbf{J} + \mathbf{u}^*c^*) = 0$, where the diffusive flux is $\mathbf{J} = -\mathbf{n} \cdot D\nabla c^*$ and $\mathbf{u}^*c^*$ is the advective flux ($\mathbf{n}$ is the unit vector normal). The velocity inlet condition was normal, uniform flow with speed $U_o$, $\mathbf{u} = U_o\mathbf{n}$ and the outlet condition on the downstream end of the domain was zero pressure $p^* = 0$. For solver stability we ramped the inlet velocity from zero to $U_o$ over short interval in time. Odour sources of strength $c^* = 1$ were introduced as constant concentration constraints from finite-width locations (Table 1), where the source profile at the origins was a smoothed top hat. The inlet and outlet odour boundary conditions were no diffusive flux normal to the boundary $-\mathbf{n} \cdot \mathbf{J} = -\mathbf{n} \cdot D\nabla c^* = 0$.

**5.1.3 Discretization, solvers, and convergence.** Lagrangian shape functions were used for weak-form discretization of the fluid velocity, pressure, and odour concentration fields. The order of the integration scheme was matched to the element order for all dependent variable (first-order here). The time-dependent solver employs an implicit backward differentiation formula (BDF) method with maximum second order schemes, balancing numerical stability and damping tendencies for our smoothly varying velocity and scalar concentration gradients. BDF methods use variable-order, variable step-size backward differentiation and are known for their stability [61, 62]. The variable step size taken by the solver was informed by a prescribed absolute tolerance for the nonlinear solver and an implicit formulation of the mesh Courant-Friedrichs-Lewy (CFL) number. The solution sequence for the resulting systems of equations was fully-coupled in all dependent variables (fluid velocity and pressure, scalar concentration) for each solver iteration using an affine invariant form of the damped Newton

method [63]. The nonlinear systems of equations were solved iteratively with specified convergence criteria using the direct PARDISO solver optimized for parallelized solutions of sparse systems of equations [64, 65]. The discretization schemes and solvers detailed above on the described mesh yielded good numerical stability and solution convergence with acceptable memory and computation requirements.

**5.1.4 Lagrangian coherent structures.**   To investigate the dynamics of the Lagrangian flow structures responsible for odour mixing we computed backwards finite-time Lyaponov exponent (FTLE) fields per [55]. The computation followed the methods described in [56], and the resulting backwards time FTLE fields provided a proxy for the attracting LCS in the flow. The integration time $T_{LCS}$ was set to 0.6 seconds based on preliminary investigations of the spatially-varying integral time scales. The temporal resolution of the LCS computation matched the resolution of the underlying velocity data (20 ms), and the initial spatial resolution the Lagrangian tracer grid was 250 $\mu$m. These integration parameters yielded well-defined FTLE fields with strong ridge lines that were in good qualitative agreement with spatial regions odour pair coalescence, suggestive of convergence in the FTLE computation.

## 5.2 Numerical methods

**5.2.1 Computing plume correlations.**   *Fourier decomposition and windowing of plumes.* To compute the correlations between a pair of plumes, a short-time Fourier transform (STFT) was applied to each plume using the `scipy.signal.stft` [66]. The STFT was computed for a specified window length using rectangular (boxcar) windows, with an overlap of half the window length, and using no padding or boundary. The application of different window shapes was implemented through the detrending function supplied to the STFT. The detrender first applied the desired window before z-scoring the result through mean-subtraction followed by scaling by its standard deviation. This detrending was performed so that the coefficients of the decomposition of two signal into trigonometric coefficients could be easily combined as in Eq 59 to yield the components of the Pearson correlation contributed by each harmonic. The conversion of Fourier coefficients to their trigonometric equivalents is completely standard so we have provided the derivation in the Supporting Information. The derivation shows that a signal $x[n]$ of length $L$ with discrete Fourier transform coefficients $X[k] = u_k + jv_k$ can be expressed in terms of sines and cosines as

$$x[n] = \sum_{k=0}^{\lfloor \frac{L}{2} \rfloor} a_k \cos\left(2\pi kn/L\right) + b_k \sin\left(2\pi kn/L\right), \tag{27a}$$

$$a_k = \begin{cases} u_k & \text{if } k \in \{0, L/2\}, \\ 2u_k & \text{otherwise.} \end{cases} \tag{27b}$$

$$b_k = -2v_k, \tag{27c}$$

where $\lfloor \cdot \rfloor$ is the floor function.

*Computing correlations from Fourier coefficients.* The component of the correlation between two plumes for a given time window in a given harmonic was computed by combining the sine and cosine coefficients from each source according to Eq 3. To compute the full Pearson correlation we summed the harmonic correlations over all harmonics as in Eq 2. To compute the distribution of correlations at a given intersource distance we pooled all correlations for all time windows computed for all pairs of sources separated by the given intersource distance. Cumulative distribution functions for some of this data are plotted in the top panels of Fig 7.

We fitt the distribution of correlations.

**5.2.2 Modeling the dependence of Fourier coefficients on intersource distance.** To compute the empirical distribution of the Fourier coefficients from two sources at a fixed intersource distance $s$ and for a given harmonic $n$, we first listed all pairs of sources that were the desired distance apart. Our source locations only differed in the $y$-coordinate, and we used a signed distance, so $\text{dist}(y_i, y_j) \triangleq y_i - y_j$. Therefore, a desired intersource distance $d$ yielded a list of ordered pairs of sources $(y_i, y_j)$ such that $y_i - y_j = d$. We then concatenated the sine and cosine coefficients in the given harmonic for all time windows across all sources in the first coordinate of each pairing. This yielded a vector of length

$$N(s) = 2 \times (\#\text{time windows per source}) \times (\#\text{ordered pairs a distance } s \text{ apart}), \quad (28)$$

where the factor of 2 is because of the concatenation of both sine and cosine coefficients. The length is a function of intersource distance $s$ because the number of pairs a given distance apart was distance-dependent. We then stacked this vector on top of the same concatenation of data applied to the sources in the second coordinate of each pairing. This yielded a $2 \times N(s)$ matrix $\mathbf{Data}_n(s)$,

$$\mathbf{Data}_n(s) = \underbrace{\begin{bmatrix} n'th \text{ Fourier coefficient from } \mathbf{first} \text{ source in all ordered pairs a distance } s \text{ apart} \\ n'th \text{ Fourier coefficient from } \mathbf{second} \text{ source in all ordered pairs a distance } s \text{ apart} \end{bmatrix}}_{N(s) \text{ elements}}, \quad (29)$$

where each element of the first row was one Fourier coefficient computed in one time window from one source, and the corresponding element in the second row was the corresponding coefficient for another source a distance $s$ apart. We plotted some of this data in Fig 6.

The columns of $\mathbf{Data}_n(s)$ represent samples from the joint distribution $p(a_n, c_n | s)$ of Fourier coefficients for the $n$'th harmonic, from sources a distance $s$ apart. To fit a zero-mean bivariate Gaussian to $\mathbf{Data}_n(s)$ as in Eq 14b we computed the principle variances $\lambda_n(s)$ and $\mu_n(s)$ along the major ($y = x$) and minor ($y = -x$) axes, respectively. To determine these we computed the variances of the projections of the data on the unit vectors $[1, 1]^T / \sqrt{2}$ and $[1, -1]^T / \sqrt{2}$, respectively,

$$\lambda_n(s) = \text{var}\left(\frac{[1, 1] \cdot \mathbf{Data}_n(s)}{\sqrt{2}}\right), \quad (30a)$$

$$\mu_n(s) = \text{var}\left(\frac{[1, -1] \cdot \mathbf{Data}_n(s)}{\sqrt{2}}\right). \quad (30b)$$

We computed the variance $\sigma_n^2$ as the mean of the principle variances at $s = 0$,

$$\sigma_n^2 = \frac{\lambda_n(0) + \mu_n(0)}{2} = \frac{\lambda_n(0)}{2}. \quad (31)$$

This is equivalent to computing the pooled variance of all coefficients from all sources for the given harmonic.

**5.2.3 Parametric fits to $\rho_n(s)$ as a function of distance.** We next fit each $\rho_n(s)$ to the parametric form in Eq 24 by non-linear least squares using `curve_fit` from the `scipy.optimize` package [66]. The parameters to be learned were the length scale $\iota$ and the offset $b$. We constrained the parameters to be non-negative, and initialized $\iota$ to 1 and $b$ to 0. All other parameters to `curve_fit` were kept at their default values. Because the data sometimes exhibited changes in length scale with intersource distance (see e.g. Fig 8B) we only used the

data for intersource distances up to one pitch to compute fit parameters. Some example fits and parameters learned are in Fig 8.

**5.2.4 Fitting the distribution of correlations.**   We fit intermittent and non-intermittent versions of three different models to the correlation data. The models differed in the probability distributions they used to fit the correlations. Each model was based on a probability distribution on non-negative values. To extend the domain to cover negative values, the same base distribution was used but applied to the underline{absolute value} of the negative values, and with new set of parameters to cover these values, and the overall distribution normalized to one. That is, for a base distribution on non-negative values with parameters $\theta_+$,

$$p_0(x|\theta_+) = \frac{1}{Z(\theta_+)} f(x, \theta_+), \quad x \in [0, \infty) \tag{32}$$

the extended distribution was defined as

$$p(x|\theta_+, \theta_-) = \frac{1}{Z(\theta_+) + Z(\theta_-)} \begin{cases} f(x, \theta_+) & x \geq 0 \\ f(|x|, \theta_-) & x < 0. \end{cases} \tag{33}$$

For example, the asymmetric Laplacian distribution in Eq 21 corresponds to a base Exponential distribution, with parameters $\lambda = (1 + \rho_n(s))/2$ covering the positive correlations, and $\mu = (1 - \rho_n(s))/2$ covering the negative correlations.

**Note:** In what follows we will frequently refer to the extended models by their base distributions, so e.g. Exponential when referring to the asymmetric Laplacian distribution.

**Evaluating models by comparing CDFs.**   We evaluated correlation models by comparing their predicted cumulative distribution functions (CDFs) to those we observed.

To compute the empirical CDF for a set of $N$ observed correlations, we first sorted the values in ascending order, to yield

$$\text{sorted data} = \{r_1, \cdots, r_N\}.$$

We then computed the empirical CDF evaluated at each data point $r_i$ as

$$F_{\text{data}}(r_i) = i/N. \tag{34}$$

The predicted CDF can be derived using the probability density function of Eq 95, which is

$$\begin{aligned} p(y|\theta) &= p(y|z = 1, \theta)p(z = 1|\theta) + p(y|z = 0, \theta)p(z = 0|\theta) \\ &= \rho(y|\theta)\iota + \eta(y|\theta)(1 - \iota), \end{aligned} \tag{35}$$

where $\rho$ and $\eta$ are the correlation and noise distributions, respectively (see Eq 99 for a particular case). The CDF is then a linear combination of the CDFs for the correlation and noise distributions:

$$F_{\text{pred}}(r_i) = F_{\text{corr}}(r_i)\iota + F_{\text{noise}}(r_i)(1 - \iota). \tag{36}$$

When comparing CDFs the $R^2$ value can be overly optimistic since both CDFs start at zero and increase monotonically to 1. Therefore, we compared CDFs more stringently by computing the largest absolute difference in their values, which is bounded below by 0 and above by 1. That is, we computed

$$\text{FitQuality} = 1 - \max_i |F_{\text{pred}}(r_i) - F_{\text{data}}(r_i)|. \tag{37}$$

When numerically computing the predicted CDFs we sometimes encountered values outside of the valid [0, 1] range. This was particularly the case when using `scipy.stats.geninvgauss` to compute the CDF of the generalized inverse Gaussian. Therefore, when computing fit quality we only used the subset of points for which the CDF values were valid.

**Fitting models using expectation-maximization.** To fit models with intermittency we had to determine which of a set of observed correlations $\{y_1, \ldots, y_N\}$ at a given harmonic and intersource distance were attributed to noise. From Eq 95 this requires determining the values of the binary latent variables $\{z_1, \ldots, z_N\}$ corresponding to each of the observations. To fit the parameters $\theta$ of a model while accounting for these latent variables, we used the Expectation-Maximization algorithm [67]. Initialized with an initial guess of the model parameters, this algorithm consists of repeatedly estimating the values of the latent variables (the 'E'-step), then using those estimates to update the estimates of the model parameters (the 'M'-step). Below, we summarize the parameter updates. Derivations are supplied in Sec S5 in S1 File.

The E-step update for all models at the $t+1$'st iteration has the simple form

$$
z_{i,t+1} = \begin{cases} 1 & \text{if } \log \dfrac{\rho(y_i|\theta_t)}{\eta(y_i|\theta_t)} > \log \dfrac{1 - \iota_t}{\iota_t}, \\[2mm] 0 & \text{otherwise.} \end{cases}
\tag{38}
$$

Here $\rho(y_i|\theta_t)$ and $\eta(y_i, \theta_t)$ are the probabilities of the observation $y_i$ under the correlation, and noise, distributions, respectively, $\theta_t$ is the current estimate of the model parameters, and $\iota_t$ is the current estimate of the intermittency parameter. The E-step therefore declares an observation $y_i$ to have been a correlation rather than noise if its probability according to the correlation distribution is greater than its probability according to the noise distribution by a threshold that depends on the current estimated intermittency level. For the non-intermittent models, all observations were marked as being correlations.

During the M-step model parameters were updated using the current estimate of the latent variables $\{z_{1,t}, z_{2,t}, \ldots, z_{N,t}\}$. All models had two parameters that related to intermittency: the intermittency level $\iota$, and the standard deviation $\sigma$ of the noise distribution. These parameters were updated the same way for all models. For the non-intermittent models, $\iota$ was fixed at 1, and $\sigma$ at 0. Otherwise, these parameters were updated as described below.

To update the intermittency level $\iota$, it is helpful to define the data intermittency at time $t$ as

$$
\overline{\iota}_t = \frac{\sum_{i=1}^{N} z_{i,t}}{N}.
\tag{39}
$$

The intermittency level is updated by balancing the data intermittency against a prior on intermittency, $\overline{\iota}_0$,

$$
\iota_t = \frac{N\alpha\overline{\iota}_0 + N\overline{\iota}_t}{N\alpha + N} = \frac{\alpha\overline{\iota}_0 + \overline{\iota}_t}{\alpha + 1}.
\tag{40}
$$

This can be interpreted as weighting $N\alpha$ observations with an intermittency of $\overline{\iota}_0$ against $N$ observations at the data intermittency $\overline{\iota}_t$. The hyperparameter $\overline{\iota}_0$ was fitted using grid search (see below). The hyperparameter $\alpha$, which sets the relative strength of the prior, was fixed at 1.

We updated the noise level by setting its variance to be the mean sum of squares of the observations attributed to noise in the E-step. Intuitively, this is because under our assumption that the noise distribution has mean zero, the mean sum of squares of the noise observations is an estimate of its variance. If the E-step did not flag any observations as noise we set the estimated noise variance to a small fraction of the variance of the observed correlations. We did this because setting the noise level to exactly zero would mean that the estimated noise

variance would remain at zero for all future iterations. This is because the probability of any non-zero observations would be 0 under this degenerate noise distribution, so all observations in the next E-step would also be flagged as correlations and not noise, leaving the noise variance stuck at zero.

Letting $N_t^0 = \sum_{i=1}^N (1 - z_{i,t})$ be the number of observations attributed to noise after the current E-step,

$$\sigma_t^2 = \begin{cases} \sum_{i=1}^N (1 - z_{i,t}) y_i^2 / N_t^0 & \text{if } N_t^0 > 0 \\ \sigma_y^2 / 1000 & \text{otherwise,} \end{cases} \tag{41}$$

where $\sigma_y^2$ is the variance of the observed correlations.

The rest of the parameters varied by model so their updates will be described for each model separately.

*Exponential model.* In addition to the intermittency parameters, the Exponential model had two parameters, $\lambda$ and $\mu$, describing the decay rate of the positive, and negative, correlations, respectively. The M-step updates for these parameters had the closed form

$$\lambda_t = \frac{1}{N_t + 10^{-8}} \left( N_t^+ \overline{|y_t^+|} + \sqrt{N_t^+ N_t^- \overline{|y_t^+||y_t^-|}} \right), \tag{42a}$$

$$\mu_t = \max \left[ \frac{1}{N_t + 10^{-8}} \left( N_t^- \overline{|y_t^-|} + \sqrt{N_t^+ N_t^- \overline{|y_t^+||y_t^-|}} \right), 10^{-6} \right]. \tag{42b}$$

Here $N_t = \sum_{i=1}^N z_{i,t}$ is the number of observations that were designated as correlations in the E-step. These can be split into $N_t^+$ positive and $N_t^-$ negative correlations, and $\overline{|y_t^+|}$ and $\overline{|y_t^-|}$ are the averages of the absolute values of the corresponding observations. The maximum operations prevents $\mu_t$ from being set to zero when the E-step did not designate any observations as noise. The added factor of $10^{-8}$ in the denominators prevents division by zero when $N_t = 0$ i.e. all observations were estimated to be noise.

*Gamma model.* The Gamma model has all the parameters of the Exponential model, plus two shape parameters, $k$ and $m$, to describe the positive and negative correlations. To update the parameters during the M-step we minimized the negative log likelihood of the observations that were marked as correlations in the E-step:

$$\{\lambda_t, \mu_t, k_t, m_t\} = \operatorname{argmin} L_t(\lambda, \mu, k, m) \tag{43a}$$

$$\begin{aligned} L_t(\lambda, \mu, k, m) &= \log \left( \Gamma(m) \mu^m + \Gamma(k) \lambda^k \right) \\ &\quad + \frac{N_t^+}{N_t + 10^{-8}} \left( \frac{\overline{|y_t^+|}}{\lambda} - (k-1) \log \overline{|y_t^+|} \right) \\ &\quad + \frac{N_t^-}{N_t + 10^{-8}} \left( \frac{\overline{|y_t^-|}}{\mu} - (m-1) \log \overline{|y_t^-|} \right). \end{aligned} \tag{43b}$$

The minimization was performed using Nelder-Mead method as implemented by the `scipy.minimize` [66] function. The bounds of the search were $[10^{-6}, 10]$ for $\lambda$ and $\mu$, and $[0, 10]$ for $k$ and $m$. The search was initialized at the parameter values from the previous iteration, clipped to lie within these bounds.

*Generalized inverse Gaussian.* The Generalized inverse Gaussian model has all the parameters of the Gamma model, plus two additional shape parameters, $\alpha$ and $\beta$, for the positive and

negative correlations, respectively. During the M-step we performed a damped update of the parameters towards the minimum of the negative log likelihood of the observations that marked as correlations in the E-step:

$$
\begin{aligned}
\{\lambda_t, \mu_t, k_t, m_t, \alpha_t, \beta_t\} \quad &= \delta \{\lambda_{t-1}, \mu_{t-1}, k_{t-1}, m_{t-1}, \alpha_{t-1}, \beta_{t-1}\} + (1-\delta) \operatorname{argmin} L_t(\lambda, \mu, k, m, \alpha, \beta) \\
L_t(\lambda, \mu, k, m, \alpha, \beta) \quad &= \log\left(2\lambda K_v(k, \alpha) + 2\mu K_v(m, \beta)\right) \\
&\quad + \frac{N_t^+}{N_t + 10^{-8}}\left((k-1)\log(\lambda) - (k-1)\log\overline{|y_t^+|} + \frac{\alpha}{2\lambda}\overline{|y_t^+|} + \frac{\alpha\lambda}{2}\overline{\frac{1}{|y_t^+|}}\right) \qquad (44a) \\
&\quad + \frac{N_t^-}{N_t + 10^{-8}}\left((m-1)\log(\mu) - (m-1)\log\overline{|y_t^-|} + \frac{\beta}{2\mu}\overline{|y_t^-|} + \frac{\beta\mu}{2}\overline{\frac{1}{|y_t^-|}}\right),
\end{aligned}
$$

where $K_v(a, b)$ is the modified Bessel function of the second kind of real order $a$ evaluated at $b$. Damping was used to aid convergence, with the damping factor $\delta$ fixed at 0.5. The minimization was performed the same way as for the Gamma model, with the new parameters $\alpha$ and $\beta$ bounded to $[10^{-6}, 10]$. The search was initialized at the parameter values from the previous iteration, clipped to lie within these bounds.

**Ancestral initialization.**   The Gamma distribution subsumes the Exponential distribution, because the former reduces to the latter when the shape parameter is set to 1. In turn, the generalized inverse Gaussian distribution subsumes the Gamma distribution, because the former reduces to the latter when its $\beta$ parameter (the coefficient of $1/x$ in the argument of the exponential) is set to zero. Because models with fewer parameters can be easier to fit, to aid the fitting process we fit our models ancestrally. That is, to fit a model using the Gamma distribution we first fit the data using an Exponential distribution, and then initialized the Gamma fit with the parameters of the Exponential, initializing the shape parameter of the Gamma to zero. Similarly, when fitting a generalized inverse Gaussian, we initialized its $\beta$ parameter at zero, and initialized the remaining parameters from those of the best fit Gamma. The Gamma in turn was initialized with the parameters of the best fit Exponential.

**Evaluating models using nested cross-validation.**   We fit a range of models to the correlation data by testing all combinations of the following hyperparameters:

- Whether the model was intermittent or not;

- If intermittent, the mean of the prior Beta distribution on the intermittency parameter, for the values $\{0.1, 0.2, \ldots, 0.9\}$.

- The base probability for the correlations, whether Exponential, Gamma, or Generalized Inverse Gamma.

  The remaining hyperparameters, listed below, were held constant at the stated values

- The minimum (0) and maximum (10) shape parameters for the Gamma and Generalized Inverse Gaussian distributions.

- The strength (1) of the Beta prior on intermittency.

We characterized these models in two ways. First, we estimated the performance of each model in fitting the correlation data. To do so, we repeatedly split the data into training (67%) and test (33%) sets. Each model was fitted to the training set and evaluated on the test set. Its performance was estimated as its average performance over three such random splits.

Secondly, we estimated the rank of each model, so that we could say e.g. which model was best overall, or which was the best among those with intermittency. One way to do this would be simply by ranking the performances we computed above. However, doing so would risk overfitting. To see why, note that by ranking the models we are in effect fitting the

hyperparameters. Ranking the models using the same data used to evaluate their performance would then mean estimating performance with the same data used to fit the (hyper)parameters, risking overfitting.

To estimate the rank of each model while avoiding making this estimation on the same data that was used to estimate performance, we performed an inner loop of cross validation for each iteration of the outer loop. In each iteration of the inner loop, the training set provided by the outer loop was further split at random into a training subset (67%) and a validation set (33%). Models for every setting of the hyperparameters were fit on the training subset, and evaluated on the validation set. Models were ranked by their mean validation performance over three such random splits. The resulting model ranks were then averaged over iterations of the outer loop.

To see why this nested cross-validation procedure avoids overfitting, observe that each iteration of the outer loop yields, first, an estimate of model performance as computed on the test data, and second, model rank as determined from the training data by the inner cross-validation loop. Therefore, in each outer loop iteration, model rankings and model performance are estimated using different data. We then reduce the noise in these estimates by averaging over iterations of the outer loop.

**5.2.5 Regressing information on frequency.** To produce the results in Fig 12 we regressed, for each value of intersource distance, Fisher information on frequency. The Fisher information at some intersource distances exhibited outliers that overly influenced the results of a standard linear regression, as judged by visual inspection. In addition to filtering out data points for which the computed Fisher information had NaN, infinite, or negative values, we took two measures to make the results more robust to these outliers. First, we limited the range of frequencies considered to be 1 Hz to 15 Hz. This was because some of the data, particularly in the supporting simulations, showed sudden drops in Fisher information above $\sim 15$ Hz. Secondly, we replaced ordinary linear regression with robust regression. We used Huber regression, as implemented in scikit-learn's `linear_model.HuberRegressor`. We used the default parameters, except for `max_iter` which we set to 10,000.

**5.2.6 Bootstrapping procedure.** To estimate the variability of the various statistics used in our approached we computed their bootstrap distributions. These statistics were computed from the trigonometric form of the Fourier decompositions computed for each plume in each time window. Therefore we first computed these coefficients for all time windows, and then created each bootstrap dataset by sampling time windows with replacement until we had as many time windows as the original dataset. We generated 50 bootstrap datasets in this way. We then computed our various statistics using the data for the time windows chosen for each bootstrap dataset. This then gave us as many point estimates of each statistics as bootstrap datasets, from which we computed e.g. the 5th, 50th and 95th percentiles of Fisher information shown in Fig 10A.

**5.2.7 Generating surrogate data.** In brief, we generated surrogate data consisting of artificial plumes with covariance structure similar to real plumes but for which we could adjust the amount of spatial information. To generate these plumes we randomly generated coefficients according to the bivariate normal model of Eq 14b using covariance kernel $K(n, s)$ to relate the coefficients from plumes a distance $s$ apart. Specifically,

$$\langle a_m c_n \rangle = \langle b_m d_n \rangle = \delta(m, n) K(n, s), \tag{45}$$

where as usual, $a_m$ and $c_n$ are the cosine coefficients for the $m$ and $n$'th frequency components, generated by two plumes a distance $s$ apart, $b_m$ and $d_n$ are the corresponding sine coefficients, and expectations are taken over time windows, and the $\delta$ function $\delta(m, n)$ ensures that

coefficients for different harmonics are uncorrelated. We combined these coefficients with their corresponding sine and cosine waveforms to generate the surrogate signals.

In detail, we generated signals for $M$ sources, each of length $2\lfloor L/2 \rfloor + 1$, by first creating the kernel relating the trigonometric coefficients of their Fourier decompositions. The signals had zero-mean, therefore for each source, we needed to specify $\lfloor L/2 \rfloor$ sine and cosine coefficients, corresponding to the normalized frequencies $\frac{1,2,...,\lfloor L/2 \rfloor}{2\lfloor L/2 \rfloor}$, covering all positive harmonics of the fundamental up to the Shannon limit. The kernel needed to specify how each harmonic at every source was correlated with each harmonic at every other source, a total of $M^2\lfloor L/2 \rfloor^2$ values. We organized these values into a symmetric kernel matrix of size $M\lfloor L/2 \rfloor \times M\lfloor L/2 \rfloor$. The elements of this matrix were determined as

$$K[Mn + i, Mn + j] = k(i, j, n),\qquad(46)$$

where $k(i, j, n)$ was the desired covariance of the trigonometric coefficients for the $n$'th harmonic for sources $i$ and $j$ (see below). We assumed coefficients at different harmonics were uncorrelated.

To compute cosine coefficients with covariance $K$, we first computed the Cholesky decomposition $K = LL^T$. We then generated $M\lfloor L/2 \rfloor$ i.i.d. samples from a standard normal, $u$, and determined the cosine coefficients as $c = Lu$. We generated a second set of samples, $v$, from the standard normal in the same way, and used those to produce the sine coefficients $s = Lv$. Indexing these coefficients by source $m$ and harmonic $n$, we produced the signals at each source by combining the coefficients with their corresponding trigonometric waveforms,

$$x_m[t] = \sum_{n=1}^{\lfloor L/2 \rfloor} c_{mn} \cos\left(\frac{\pi n}{\lfloor L/2 \rfloor}t\right) + s_{mn} \sin\left(\frac{\pi n}{\lfloor L/2 \rfloor}t\right), \quad t \in 0, 1, \cdots, 2\lfloor L/2 \rfloor.\qquad(47)$$

**Kernel functions.** We used several different kernel functions $k(i, j, n)$ to generate the surrogate data in the text. Each of these kernel functions was the product of a term $S(n)$ determining the power at the $n$'th harmonic, and a function $G(i, j, n)$ determining the correlation between sources $i$ and $j$, that in some cases depended on the harmonic. In every case, this correlation function was a function only of the absolute difference $s = |i - j|$ of the sources, so it simplified to $G(s, n)$. Thus our kernels were of the form

$$k(i, j, n) = G(|i - j|, n)S(n).\qquad(48)$$

Our kernels differed by whether they used a flat power spectrum or one similar to that of the simulations. We captured both cases by using a $1/f$ power spectrum:

$$P(n, \alpha) = \max\left(\frac{n}{2\lfloor L/2 \rfloor + 1}f_s, 1\right)^{-\alpha},\qquad(49)$$

where $f_s$ was the sample rate in Hz. The maximum operation provides a frequency cutoff at 1 Hz below which the power is set to 1. To achieve a 'white' spectrum, we set $\alpha = 0$. To achieve a 'pink' spectrum similar to our CFD simulations, we set $\alpha = 4$.

Our surrogate data also differed in the correlation functions used. For the data where all frequencies were equally informative, we set

$$G(|i - j|, n) = G(|i - j|) = 2\exp\left(-|i - j|/12\right) - 1.\qquad(50)$$

For the data where the high frequencies were more informative for sources close together than

the lower frequencies, we set

$$G(|i-j|, n) = \exp\left(-|i-j|/R(n)\right), \quad R(n) = \begin{cases} 12 & n < \lfloor L/2 \rfloor/2, \\ 2 & \text{otherwise.} \end{cases} \tag{51}$$

Thus the length scale of decay was 6 times shorter for high harmonics (those in the upper half of the range) than for low harmonics.

The table below summarizes the surrogate datasets and the power and correlation functions used in each.

**5.2.8 Computing integral length scales.** We computed integral length scales from velocity autocorrelation functions as defined in [68]. For example, to compute the integral length scale in the $y$-direction at a location $(x, y)$ of interest, we first computed the $y$-velocity autocorrelation function in the $y$ direction at that location by computing the time average of the product of the $y$-velocity at the location of interest, $u_y(t;x, y)$ and that at a fixed positive displacement $r$ in the $y$-direction, $u_y(t;x, y + r)$. That is, we computed

$$Q_{yy}^+(r; x, y) = \langle u_y(t; x, y) u_y(t; x, y + r) \rangle_t. \tag{52}$$

We also computed the symmetric average in the negative $y$-direction,

$$Q_{yy}^-(r; x, y) = \langle u_y(t; x, y) u_y(t; x, y - r) \rangle_t. \tag{53}$$

We then averaged these two to arrive at the autocorrelation for a single, positive value of $r$

$$Q_{yy}(r; x, y) = \frac{Q_{yy}^+(r; x, y) + Q_{yy}^-(r; x, y)}{2}. \tag{54}$$

If e.g. $(x, y + r)$ was not in the domain, then $Q_{yy}^+$ would not be computed and $Q_{yy}$ took the value of $Q_{yy}^-$, and similarly if $(x, y - r)$ was not in the domain.

We then computed the longitudinal velocity correlation function by normalizing the velocity correlation function by its value at $r = 0$

$$f_y(r; x, y) = \frac{Q_{yy}(r; x, y)}{Q_{yy}(0; x, y)}. \tag{55}$$

Finally, we computed the integral length scale by integrating this function from 0 to $\infty$:

$$L_U(x, y) = \int_0^\infty f(r; x, y) \, dr. \tag{56}$$

The $r$ values we used were discrete and spaced $\Delta r = 0.02\phi$ apart, so we approximated the integral above as

$$L_U \approx \sum_{n=0}^{N-1} f(n\Delta r; x, y)\Delta r, \tag{57}$$

where $N$ was the number of $r$ values used.

In S20 Fig we have plotted the longitudinal velocity autocorrelation functions in the $x$ and $y$ directions, and the corresponding integral length scales, for two locations of interest.

### 5.3 Analytical methods

**5.3.1 Decomposing Pearson correlation.** We can express two concentration profiles $x(t)$ and $y(t)$ observed over a window of width $T$ in terms of their Fourier decompositions,

$$x(t) = \sum_{n=0}^{\infty} \tilde{a}_n \sin\left(2\pi n t/T\right) + \tilde{b}_n \cos\left(2\pi n t/T\right) \tag{58a}$$

$$y(t) = \sum_{n=0}^{\infty} \tilde{c}_n \sin\left(2\pi n t/T\right) + \tilde{d}_n \cos\left(2\pi n t/T\right). \tag{58b}$$

Using the orthogonality of different harmonics of the fundamental frequency we can express the Pearson correlation in terms of the Fourier components as

$$r = \frac{1}{T} \frac{\int_0^T (x(t) - \overline{x})(y(t) - \overline{y})\, dt}{\sigma_x \sigma_y} = \sum_{n=1}^{\infty} \frac{\tilde{a}_n \tilde{c}_n + \tilde{b}_n \tilde{d}_n}{2\sigma_x \sigma_y} = \frac{1}{2} \sum_{n=1}^{\infty} a_n c_n + b_n d_n = \sum_n r_n. \tag{59}$$

**5.3.2 Relating the coefficients at one source to those at another.** To determine the uncertainty about the coefficients $c_n$, $d_n$ of a component waveform at a second source,

$$y_n(t) = c_n \cos\left(\omega t\right) + d_n \sin\left(\omega t\right), \tag{60}$$

given the coefficients $a_n$, $b_n$ of a first,

$$x_n(t) = a_n \cos\left(\omega t\right) + b_n \sin\left(\omega t\right), \tag{61}$$

we express the former as the sum of a deterministic component formed of a scaled and phase-shifted version of the component waveform from the first source, plus noisy residual. That is,

$$y_n(t) = \beta_n[a_n \cos\left(\omega t + \theta_n\right) + b_n \sin\left(\omega t + \theta_n\right)] + \varepsilon(t), \tag{62}$$

where $\beta_n$ and $\theta_n$ are the best-fit scaling and phase-shift of the first source. These can be found to satisfy (see below)

$$\langle a_n c_n \rangle = \langle b_n d_n \rangle = \sigma^2 \beta_n \cos\left(\theta_n\right), \tag{63a}$$

$$\langle b_n c_n \rangle = -\langle a_n d_n \rangle = \sigma^2 \beta_n \sin\left(\theta_n\right), \tag{63b}$$

where as usual expectations are over time windows.

The residual $\varepsilon(t)$ therefore captures the uncertainty remaining about the second component waveform, given knowledge of the first. To determine its statistical properties we express it by subtracting Eq 62 from Eq 61 as

$$\varepsilon(t) = c_n \cos\left(\omega t\right) + d_n \sin\left(\omega t\right) - \beta_n[a_n \cos\left(\omega t + \theta_n\right) + b_n \sin\left(\omega t + \theta_n\right)]. \tag{64}$$

This is a linear combination of sinusoidal waveforms at the same frequency $\omega$, so we can write it as

$$\varepsilon(t) = \tilde{c}_n \cos\left(\omega t\right) + \tilde{d}_n \sin\left(\omega t\right). \tag{65}$$

By performing trigonometric expansions, the coefficients are found to be

$$\tilde{c}_n = c_n - \beta_n [a_n \cos(\theta_n) + b_n \sin(\theta_n)], \tag{66a}$$

$$\tilde{d}_n = d_n - \beta_n [b_n \cos(\theta_n) - a_n \sin(\theta_n)]. \tag{66b}$$

For $n > 0$, $a_n$, $b_n$, $c_n$, and $d_n$ have mean zero (Eq 8). Therefore the coefficients above, and in turn $\varepsilon(t)$, as linear combinations of zero-mean random variables, are also zero mean.

To determine the variance of the residuals, we first show in Supporting Information Sec S6.2 in S1 File that the coefficients $\tilde{c}_n$ and $\tilde{d}_n$ are uncorrelated. Therefore, from Eq 65, the residual variance is the scaled sum of the variances of the coefficients. To compute these variances, we have

$$
\begin{aligned}
\mathrm{var}(\tilde{c}_n) &= \langle (c_n - \beta_n [a_n \cos(\theta_n) + b_n \sin(\theta_n)])^2 \rangle \\
&= \langle c_n^2 \rangle + \beta_n^2 \cos(\theta_n)^2 \langle a_n^2 \rangle + \beta_n^2 \sin(\theta_n)^2 \langle b_n^2 \rangle - 2\beta_n \cos(\theta_n)\langle a_n c_n \rangle - 2\beta_n \sin(\theta_n)\langle b_n c_n \rangle \\
&= \sigma_n^2 + \beta_n^2 \sigma_n^2 - 2\beta_n \cos(\theta_n)\langle a_n c_n \rangle - 2\beta_n \sin(\theta_n)\langle b_n c_n \rangle \\
&= \sigma_n^2 + \beta_n^2 \sigma_n^2 - 2\sigma_n^2 \beta_n^2 \cos(\theta_n)^2 - 2\sigma_n^2 \beta_n^2 \sin(\theta_n)^2 \\
&= \sigma_n^2 - \beta_n^2 \sigma_n^2
\end{aligned}
\tag{67}
$$

$$= \mathrm{var}(\tilde{d}_n), \tag{68}$$

where we've used the relations in Eq 66 to express coefficient correlations in terms of $\beta_n$ and $\theta_n$, and the final equality follows by temporal stationarity. We then arrive at

$$\mathrm{var}(\varepsilon) = \mathrm{var}(\tilde{c}_n)\langle \cos(\omega t)^2 \rangle + \mathrm{var}(\tilde{d}_n)\langle \sin(\omega t)^2 \rangle = \sigma_n^2(1 - \beta_n^2) \triangleq \eta_n^2. \tag{69}$$

Knowledge of the mean and variance would completely specify the probability distribution of the residual if it were a Gaussian random variable. Although it can be expressed as a linear combination of the Gaussian random variables $a_n$, $b_n$, $c_n$ and $d_n$ (Eq 64), these latter variables are not necessarily independent. Therefore, their combination is not necessarily Gaussian. To enforce this requirement we made the assumption 2.2.

To determine the scaling $\beta_n$ and phase-shift $\theta_n$, we equate Eq 62 an Eq 62 and match coefficients of $\cos(\omega t)$ and $\sin(\omega t)$, to get

$$c_n = \beta_n [a_n \cos(\theta_n) + b_n \sin(\theta_n)] \tag{70a}$$

$$d_n = \beta_n [b_n \cos(\theta_n) - a_n \sin(\theta_n)]. \tag{70b}$$

Correlating these equations against $a_n$ and $b_n$, respectively, and using Eq S29 in S1 File and the definition Eq 10 of the coefficient variances $\sigma_n^2$, we get

$$\langle a_n c_n \rangle = \beta_n \langle a_n^2 \rangle \cos(\theta_n) = \sigma_n^2 \beta_n \cos(\theta_n), \tag{71a}$$

$$\langle b_n d_n \rangle = \beta_n \langle b_n^2 \rangle \cos(\theta_n) = \sigma_n^2 \beta_n \cos(\theta_n). \tag{71b}$$

Correlating against $b_n$ and $a_n$, respectively, and using Eq S30 in S1 File, we get

$$\langle b_n c_n \rangle = \beta_n \langle b_n^2 \rangle \sin(\theta_n) = \sigma_n^2 \beta_n \sin(\theta_n), \tag{72a}$$

$$\langle a_n d_n \rangle = -\beta_n \langle a_n^2 \rangle \sin(\theta_n) = -\sigma_n^2 \beta_n \sin(\theta_n). \tag{72b}$$

**5.3.3 Deriving the distribution of correlations.** Here we derive the distribution $p(r_n|s)$ of the component correlations $r_n$ for the $n$'th harmonic for sources separated by a distance $s$, given our assumptions about the distribution of Fourier coefficients, $a_n$ and $b_n$, from the first source and $c_n$ and $d_n$ from the second source.

**Accounting for location dependence.** We must first account for the fact that the distribution of correlations is formed by the contribution of all pairs of sources separated by the given distance. To do so, we first assign integer indices to the sources. A pair of sources is then $(i, j)$, the distance between them, $D_{ij}$, and the number of pairs a distance $s$ apart, $N_s$. We can then use the rules of probability to express the distribution of correlations in terms of the contributions from each pair

$$p(r_n|s) = \sum_{(i,j)} p((i,j)|s) p(r_n|(i,j), s). \tag{73}$$

The first term in the summand is the probability of the pair $(i, j)$ being selected given the sources are a distance $s$ apart. Because pairs at this distance will be selected uniformly and all others will not,

$$p((i,j)|s) = \begin{cases} 1/N_s & \text{if } D_{ij} = s, \\ 0 & \text{otherwise.} \end{cases} \tag{74}$$

Our decomposition then simplifies to

$$p(r_n|s) = \frac{1}{N_s} \sum_{(i,j):D_{ij}=s} p(r_n|(i,j), s). \tag{75}$$

The summand is the distribution of correlations, given the pair of sources generating them and the distance between those sources. However, once the pair generating the correlations is specified the distance between the pair is redundant and we can remove it from the conditioners, yielding

$$p(r_n|s) = \frac{1}{N_s} \sum_{(i,j):D_{ij}=s} p(r_n|(i,j)). \tag{76}$$

The new summand is the distribution of correlations generated by a given pair of sources. This in turn is related to the Fourier coefficients from each source. Let $a_n$, $b_n$ be the cosine and sine coefficients from the first source in the pair, and $c_n$, $d_n$ the corresponding coefficients from the second source. Then

$$p(r_n|(i,j)) = \int p(r_n|a_n, b_n, c_n, d_n) \, p(a_n, b_n, c_n, d_n|(i,j)) \, da_n db_n dc_n dd_n. \tag{77}$$

The distribution of correlations for a given intersource distance is then

$$p(r_n|s) = \frac{1}{N_s} \sum_{(i,j):D_{ij}=s} \int p(r_n|a_n, b_n, c_n, d_n) \, p(a_n, b_n, c_n, d_n|(i,j)) \, da_n db_n dc_n dd_n. \qquad (78)$$

This expression depends explicitly on the identity of the sources, because the joint distribution of coefficients generated by each pair of sources can be different for each pair. To emphasize the relationship of coefficients from one source to those from the other we decompose the joint distribution as,

$$p(a_n, b_n, c_n, d_n|(i,j)) = p(a_n, b_n|i) \, p(c_n, d_n|a_n, b_n, (i,j)). \qquad (79)$$

To quantify effects that depend only on the relative location of sources, rather than the specific locations of specific sources we made location our independence assumptions. The first was that the distribution of coefficients from a source was the same for all sources, so

$$p(a_n, b_n|i) = p(a_n, b_n). \qquad (80)$$

The second was that the distribution of coefficients $c_n, d_n$ from source $j$ given the coefficients $a_n, b_n$ from source $i$ depends only on the distance between them,

$$p(c_n, d_n|a_n, b_n, (i,j)) = p(c_n, d_n|a_n, b_n, D_{ij}). \qquad (81)$$

Substituting these two assumptions into Eq 79, and that result into Eq 79 yields Eq 5.

**Deriving the distribution of correlations.** Dropping the $n$ subscripts and the conditioning on $s$ for clarity,

$$p(r|s) = \int \delta(r - \frac{1}{2} ac - \frac{1}{2} bd) p(c, d|a, b) p(a, b) \, da \, db \, dc \, dd. \qquad (82)$$

From Eq 12 We have that

$$p(c, d|a, b) = p(c|a, b) p(d|a, b) = \mathcal{N}(c; \beta a \cos(\phi) + \beta b \sin(\phi), \eta^2) \mathcal{N}(d; -\beta a \sin(\phi) + \beta b \cos(\phi), \eta^2) \quad (83)$$

By defining $\mathbf{z} = (a, b)$, $\mathbf{x} = (c, d)$ we can write Eq 83 as

$$p(\mathbf{x}|\mathbf{z}) = \mathcal{N}(\mathbf{x}; \beta R_\phi \mathbf{z}, \eta^2), \quad R_\phi \triangleq \begin{bmatrix} \cos(\phi) & \sin(\phi) \\ -\sin(\phi) & \cos(\phi) \end{bmatrix}. \qquad (84)$$

Combining this with Eq 10 to get $p(\mathbf{z})$ we have

$$p(r|s) = \int d\mathbf{x} d\mathbf{z} \, \delta(r - \frac{1}{2} \mathbf{z} \cdot \mathbf{x}) \, p(\mathbf{z}) p(\mathbf{x}|\mathbf{z}) = \int d\mathbf{z} \, \mathcal{N}(\mathbf{z}; 0, \sigma^2) \int d\mathbf{x} \, \delta(r - \frac{1}{2} \mathbf{z} \cdot \mathbf{x}) \, \mathcal{N}(\mathbf{x}; \beta R_\phi \mathbf{z}, \eta^2). \quad (85)$$

where we've used Eq 10 to specify $p(\mathbf{z})$. The integrand in Eq 85 is rotationally invariant in $\mathbf{z}$ since any rotation in $\mathbf{z}$ is matched by $\mathbf{x}$ through the $\delta$ function. So we can compute Eq 85 by computing it for one radial slice of $\mathbf{z}$, say $\mathbf{z} = z \, \mathbf{e}_1 = (z, 0)$, and multiplying the result by $2\pi$. Switching $\mathbf{z}$ to polar coordinates $\mathbf{z} = (z, \theta)$, for which

$$p(\mathbf{z}) \, d\mathbf{z} = (2\pi\sigma^2)^{-1} \exp(-z^2/2\sigma^2) z dz d\theta \triangleq q(z) z dz d\theta, \qquad (86)$$

we have

$$p(r|s) = 2\pi \int_0^\infty dz \, q(z) z \int d\mathbf{x} \, \delta(r - \frac{1}{2} \mathbf{x} \cdot \mathbf{z}) \, \mathcal{N}(\mathbf{x}; \beta R_\phi \mathbf{z}, \eta^2). \qquad (87)$$

To compute the latter integral, we switch $\mathbf{x}$ from $(c, d)$ to $(u, t)$ coordinates, defined for $\mathbf{z} = (z, 0)$ as

$$\mathbf{x}(u, t) = \begin{bmatrix} 2u/z \\ t \end{bmatrix}, \quad d\mathbf{x} = \begin{bmatrix} dc \\ dd \end{bmatrix} = \begin{bmatrix} 2z^{-1} & 0 \\ 0 & 1 \end{bmatrix} \begin{bmatrix} du \\ dt \end{bmatrix}. \tag{88}$$

Then the area element $dcdd$ becomes $2z^{-1}dudt$ and we have for the latter integral in Eq 87,

$$\int d\mathbf{x}(c, d) \ \delta(r - \tfrac{1}{2}\mathbf{x} \cdot \mathbf{z}) \ \mathcal{N}(\mathbf{x}; \beta R_\phi \mathbf{z}, \eta^2) = 2z^{-1} \int d\mathbf{x}(u, t) \ \delta(r - \tfrac{1}{2}\mathbf{x} \cdot \mathbf{z}) \ \mathcal{N}(\mathbf{x}; \beta R_\phi \mathbf{z}, \eta^2) \tag{89}$$

$$= 2z^{-1} \int dt \int du \ \delta(r - u) \ \mathcal{N}(\mathbf{x}; \beta R_\phi \mathbf{z}, \eta^2) \tag{90}$$

$$= 2z^{-1} \int dt \ \mathcal{N}(\mathbf{x}(r, t); \beta R_\phi \mathbf{z}, \eta^2). \tag{91}$$

Now for our slice $\mathbf{z} = (z, 0)$, $\mathbf{x}(r, t) - \beta R_\phi \mathbf{z} = \left(\frac{2r}{z} - \beta z \cos(\phi), t + \beta z \sin(\phi)\right)$, so

$$\mathcal{N}(\mathbf{x}(r, t); \beta R_\phi \mathbf{z}, \eta^2) = \frac{e^{-\frac{1}{2\eta^2}\left[\left(\frac{2r}{z} - \beta z \cos \phi\right)^2 + (t + \beta z \sin(\phi))^2\right]}}{2\pi\eta^2} = \mathcal{N}(2r/z; \beta z \cos \phi, \eta^2)\mathcal{N}(t; -\beta z \sin \phi, \eta^2). \tag{92}$$

Hence Eq 91 evaluates to

$$\frac{2}{z}\int dt \ \mathcal{N}(\mathbf{x}(r, t); \beta R_\phi \mathbf{z}, \eta^2) = \frac{2}{z}\mathcal{N}(2r/z; \beta z \cos \phi, \eta^2)\int dt \ \mathcal{N}(t; -\beta z \sin \phi, \eta^2) = \frac{2}{z}\mathcal{N}(2r/z; \beta z \cos \phi, \eta^2). \tag{93}$$

Substituting this in for the latter integral in Eq 87, we have

$$p(r|s) = 4\pi \int_0^\infty dz \ \mathcal{N}(z; 0, \sigma^2) \ \mathcal{N}(2r/z; \beta z \cos \phi, \eta^2) = \frac{\exp\left[\frac{2r}{\eta^2}\left(\beta \cos(\phi) - \text{sgn}(r)\sqrt{\beta^2 \cos(\phi)^2 + \frac{\eta^2}{\sigma^2}}\right)\right]}{\sigma^2\sqrt{\beta^2 \cos(\phi)^2 + \frac{\eta^2}{\sigma^2}}}.$$

Finally, to arrive at Eq 19, we simplify the length constant of the exponential by relating $\eta^2$ in Eq 69 to $Z$ in Eq 20 as $\eta^2 = -\sigma^{-2}Z^2 - \sigma^2\beta^2\cos(\phi)^2$, and use the definition of $\rho$ in Eq 15 to get

$$\frac{-\eta^2}{\beta \cos(\phi) - \text{sgn}(r)\sqrt{\beta^2 \cos(\phi)^2 + \frac{\eta^2}{\sigma^2}}} = \frac{\sigma^2\rho^2 - \sigma^{-2}Z^2}{\rho - \text{sgn}(r)\sigma^{-2}Z} = \frac{\sigma^4\rho^2 - Z^2}{\sigma^2\rho - \text{sgn}(r)Z} = \text{sgn}(r)(Z + \text{sgn}(r)\sigma^2\rho). \tag{94}$$

### 5.3.4 Improving the model of correlations.

Our correlation model captures intermittency by introducing a binary random variable $z$ that determined whether the observation $y_n$ was of a correlation $r_n$, or noise $w$. That is,

$$y_n|z, r_n, w = zr_n + (1 - z)w, \tag{95}$$

so that when the masking variable $z = 1$, we observe a 'true' correlation $r_n$, and when $z = 0$, we observe an instance of noise instead. We assume that the masking variable $z$ obeys a Bernoulli distribution,

$$z \sim \text{Ber}(\iota_n(s)). \tag{96}$$

The intermittency parameter $\iota_n(s) \in [0, 1]$ determines the fraction of observations we deem to be true correlations, and can depend both on the harmonic $n$ and the intersource distance $s$. The true correlations $r_n$ are drawn from the asymmetric Laplacian distribution of correlations in Eq 21,

$$r_n \sim p(r_n|s), \tag{97}$$

while the noisy observations $w$ come from a zero-mean Gaussian

$$w \sim \mathcal{N}(w; 0, v_n(s)^2). \tag{98}$$

Like the intermittency parameter, the variance $v_n^2(s)$ of the noise can depend on both the harmonic and intersource distance. The predicted distribution of correlations for our 'intermittent asymmetric Laplacian' model is then a linear combination of the 'true' correlation distribution $p(r_n|s)$ and the noise distribution determined by the intermittency parameter $\iota_n(s)$

$$q(r_n|s) = \iota_n(s)\, p(r_n|s) + (1 - \iota_n(s))\, \mathcal{N}(r_n; 0, v_n^2(s)). \tag{99}$$

Setting $\iota_n(s)$ to 1 corresponds to our initial asymmetric Laplacian model, while setting it zero represents a fully intermittent case where all correlations are in fact noise. The amount of intermittency and the variance of the noise were determined from the data (see Methods, Sec 5.2.4).

To demonstrate the effect of including intermittency in our model we have added the CDFs for the distributions in Eq 99 fitted to the data in Fig 7A–7C to the corresponding panels in those plots (light blue traces), revealing an improved qualitative agreement. In Fig 7D–7F we have added the deviations between the data CDFs and those of the new model that includes intermittency. The panels show that both the positive and negative deviations have been reduced, as expected, but not eliminated.

The fact that both positive and negative deviations remain in Fig 7D–7F indicates that the data have more small values than even augmentation with intermittency can capture. Therefore, we extended our model by replacing the Exponential distributions in Eq 21 that constitute the positive and negative halves of the asymmetric Laplacian with distributions that had higher densities near zero, but that included the Exponential as special cases. One candidate is the Gamma distribution, $p(r) \propto r^{k-1} e^{r/\lambda}$. Just like the Exponential distribution, the Gamma distribution has a scale parameter, $\lambda$ for the positive correlations above. In addition, it introduces a shape parameter, $k > 0$. When $k = 1$, the Gamma distribution reduces to the Exponential distribution. When $k < 1$, the density approaches infinity as $r \to 0$. This singularity at 0 aids in modeling the large number of low correlation values we observe. Analogously to Eq 21, we 'sandwich' two Gamma distributions together so that we can also cover negative values (see also Methods Sec 5.2.4), and arrive at our correlation distribution

$$p(r_n|s) = \frac{1}{Z(k,\lambda) + Z(m,\mu)} \begin{cases} r_n^{k-1} e^{-r_n/\lambda} & r_n \geq 0, \\ |r_n|^{m-1} e^{-|r_n|/\mu} & r_n < 0. \end{cases} \tag{100}$$

Here $Z$ is the normalizing function of the Gamma distribution, $k$ and $\lambda$ are the shape and scale parameters for the positive correlations, and $m$ and $\mu$ are the corresponding parameters for the negative correlations. We have omitted the dependence of these parameters on the harmonic $n$ and the intersource distance $s$ for clarity.

Another probability distribution, which includes the Gamma (and therefore the Exponential) as a special case, is the generalized inverse Gaussian. The parameterization of this distribution that we use gives the probability density function $p(r) \propto r^{k-1} e^{-\alpha(r/\lambda + \lambda/r)/2}$. This distribution

introduces a further shape parameter $\alpha > 0$ and a term proportional to $1/r$ in the argument of the exponential that provide additional flexibility in modeling small observations. The scale of the distribution can be taken to be the length constant of the $r$ term in the exponential, $2\lambda/\alpha$. Keeping this constant while reducing $\alpha\lambda$ towards zero, one approaches the Gamma distribution. As before, we sandwich two such distributions together to arrive at our distribution of correlations,

$$p(r_n|s) = \frac{1}{Z(k,\lambda,\alpha) + Z(m,\mu,\beta)} \begin{cases} r_n^{k-1} e^{-\alpha(r_n/\lambda + \lambda/r_n)/2} & r_n \geq 0, \\ |r_n|^{m-1} e^{-\beta(r_n/\mu + \mu/r_n)/2} & r_n < 0. \end{cases} \tag{101}$$

Here $Z$ is the normalizing function of the generalized inverse Gaussian distribution, $k$, $\lambda$, and $\alpha$ are the parameters for the positive correlations, and $k$ and $\mu$ and $\beta$ are the corresponding parameters for the negative correlations. As before, we have omitted the dependence of these parameters on the harmonic $n$ and the intersource distance $s$ for clarity.

Our elaborated models of the correlations were thus of the same form as Eq 99, with $p(r_n|s)$ set to either asymmetric Laplacian, or sandwiched versions of the Gamma, or generalized inverse Gaussian distributions.

**5.3.5 Fisher information.** To compute Fisher information for the distribution in Eq 19, we need (dropping subscripts for clarity),

$$\log p(r|s) = -\log(Z) - 2\frac{|r|}{Z + \text{sgn}(r)\sigma^2\rho}.$$

Then, defining $\lambda = \sigma^2\rho$

$$\frac{\partial \log p(r|s)}{\partial s} = -\frac{Z'}{Z} + 2|r|\frac{Z' + \text{sgn}(r)\lambda'}{(Z + \text{sgn}(r)\lambda)^2} \tag{102}$$

Now defining $\lambda_\perp = \sigma^2\rho_\perp$, we have $Z' = \lambda_\perp\lambda_\perp'/Z^3$ so

$$\frac{\partial \log p(r|s)}{\partial s} = -\frac{\lambda_\perp\lambda_\perp'}{Z^4} + 2\frac{\lambda_\perp\lambda_\perp' + Z^3\text{sgn}(r)\lambda'}{Z^3(Z + \text{sgn}(r)\lambda)^2}|r| \triangleq \alpha + \beta_{\text{sgn}(r)}|r|. \tag{103}$$

Then the square is

$$\left(\frac{\partial \log p(r|s)}{\partial s}\right)^2 = \alpha^2 + 2\alpha|r|\beta_{\text{sgn}(r)} + r^2\beta_{\text{sgn}(r)}^2. \tag{104}$$

We need to compute the expectation of this with respect to $r$ under $p(r|s)$. We can split this into the contributions from each component.

**Constant term.** The expectation of the constant term is just

$$\alpha^2 = \frac{\lambda_\perp\lambda_\perp'}{Z^4} = \frac{1}{Z^2}\left(\frac{\lambda_\perp\lambda_\perp'}{Z^3}\right)^2 = \left[\frac{Z'}{Z}\right]^2. \tag{105}$$

**Linear term.** Defining $I_1(\tau) \triangleq \int_0^\infty r \exp(-2r/\tau) \, dr = \frac{\tau^2}{4}$ we have

$$
\begin{aligned}
\langle 2\alpha\beta_{\text{sgn}(r)}|r|\rangle_r \quad &= 2\alpha\beta_+ \langle |r| \rangle_{r \geq 0} + 2\alpha\beta_- \langle |r| \rangle_{r < 0} = 2\alpha \frac{\beta_+ I_1(Z+\lambda) + \beta_- I_1(Z-\lambda)}{Z} \\
&= -\frac{2\lambda_\perp \lambda_\perp' (\lambda_\perp \lambda_\perp' + \lambda Z^2 \lambda')}{Z^7} = -\frac{2}{Z}\frac{\lambda_\perp \lambda_\perp'}{Z^3}\left(\frac{\lambda_\perp \lambda_\perp'}{Z^3} + \frac{\lambda\lambda'}{Z}\right) = -\frac{2}{Z}Z'\left(Z' + \frac{\lambda\lambda'}{Z}\right).
\end{aligned}
\tag{106}
$$

**Quadratic term.** Defining $I_2(\tau) \triangleq \int_0^\infty r^2 \exp(-2r/\tau) \, dr = \frac{\tau^3}{4}$ we have

$$
\langle \beta_{\text{sgn}(r)}^2 r^2 \rangle = \beta_+^2 \langle r^2 \rangle_{r \geq 0} + \beta_-^2 \langle r^2 \rangle_{r < 0} = \frac{\beta_+^2 I_2(Z+\lambda) + \beta_-^2 I_2(Z-\lambda)}{Z}
\tag{107}
$$

$$
= \frac{1}{Z}\left[\frac{\left(\frac{\lambda_\perp \lambda_\perp'}{Z^3} + \lambda'\right)^2}{Z+\lambda} + \frac{\left(\frac{\lambda_\perp \lambda_\perp'}{Z^3} - \lambda'\right)^2}{Z-\lambda}\right] = \frac{1}{Z}\left[\frac{(Z'+\lambda')^2}{Z+\lambda} + \frac{(Z'-\lambda')^2}{Z-\lambda}\right].
\tag{108}
$$

**Fisher Information.** Collecting terms, we arrive at

$$
\mathcal{I}(s) = \left[\frac{Z'}{Z}\right]^2 - 2\frac{Z'}{Z}\left[Z' + \frac{\lambda\lambda'}{Z}\right] + \frac{1}{Z}\left[\frac{(Z'+\lambda')^2}{Z+\lambda} + \frac{(Z'-\lambda')^2}{Z-\lambda}\right].
\tag{109}
$$

This expression simplifies significantly if we assume that $\lambda_\perp(s) \approx 0$, since then $Z \approx \sigma^2$, $Z' \approx 0$, and

$$
\mathcal{I}(s) \approx \frac{1}{\sigma^2}\left[\frac{\lambda'^2}{\sigma^2+\lambda} + \frac{\lambda'^2}{\sigma^2-\lambda}\right] = \sigma^2 \rho'^2 \left(\frac{1}{\sigma^2+\sigma^2\rho} + \frac{1}{\sigma^2-\sigma^2\rho}\right) = \frac{2\rho'^2}{1-\rho^2},
\tag{110}
$$

which is Eq 23. For $\rho = (1-b)e^{-s/\gamma} + b$ as in Eq 24 we get, in terms of normalized distances $\hat{s} = s/\gamma$,

$$
\mathcal{I}(\hat{s}) = \frac{2}{\gamma^2}\frac{(1-b)e^{-2\hat{s}}}{(1+b+(1-b)e^{-\hat{s}})(1-e^{-\hat{s}})},
\tag{111}
$$

which is Eq 26.

## Supporting information

**S1 File. Supplementary text.**
(PDF)

**S1 Fig. Time-averaged statistics of simulated flow field.** (**A**) horizontal (top) and vertical (bottom) components of the mean velocity (left) and integral length scale (right) over the simulated plume domain. (**B**) mean turbulent isotropy (top) and turbulent kinetic energy (bottom) over the simulated plume domain.
(TIF)

**S2 Fig. Location independence.** $p$-values of the energy test measuring the difference between the distribution of coefficients at one source ('First source') and those at another, at each frequency. Each column corresponds to the comparison for one pair of sources.
(TIF)

**S3 Fig. Joint distribution of coefficients.** *p*-values of the energy test measuring the difference between the distributions of coefficients from all pairs of sources at a given intersouce distance (indicated on the x-axis), for each frequency. Each column corresponds to the comparison of one pair of sources. There were multiple pairs of odour sources at each intersource distance, indicated by the sizes of each block. Colours as in S2 Fig.
(TIF)

**S4 Fig. Gaussian coefficients.** *p*-values of the energy test measuring the difference between the distribution of coefficients at each source to those of a bivariate Gaussian with the same mean and covariance, at each frequency. Colours as in S2 Fig, white indicates $p < 0.001$.
(TIF)

**S5 Fig. Stationarity.** (**A**) *p*-values for the energy test comparing the distribution of sine coefficients to that of the cosine coefficients, at each source and frequency. (**B,C**) *p*-values of the Wilcoxon signed-rank test for whether the medians of the distributions of the sine, and cosine, coefficients is zero, for the data from each each source and each frequency. (**D**) *p*-values of the Wilcoxon signed-rank test for whether the median of the product of the sine and cosine coefficients is zero. Colours as in S2 Fig, white indicates $p < 0.001$.
(TIF)

**S6 Fig. Conditional Gaussianity.** *p*-values of the energy test measuring the difference between the conditional distribution of coefficients at one source given those at another (the 'conditioning source'), and a bivariate Gaussian with the same mean and covariance, for each frequency. Each column corresponds to the comparison for one pair of sources. Colours as in S2 Fig, white indicates $p < 0.001$.
(TIF)

**S7 Fig. Frequency decomposition of correlations for the Main Text data, using a Kaiser-16 window.** Compare to Fig 9.
(TIF)

**S8 Fig. Fisher information vs. frequency for the Main Text data, using a Kaiser-16 window.** Compare to Fig 12.
(TIF)

**S9 Fig. Description of the second set of computational fluid dynamics simulations.** (**A**) Domain schematic and simulation parameters. Flow was from left-to-right. Vorticity was introduced into the flow by twelve cylindrical obstacles, vertically spaced evenly at a horizontal distance of 20 cm from the inlet (see also panel B). Point odour sources were placed at various vertical locations on the dashed line 40 cm horizontally from the flow inlet. Flow over these sources carried the odours to downstream probes. (**B**) Details of the obstacles and the simulation mesh. The cylindrical obstacles were 38 mm in diameter and evenly spaced 38 mm apart. The resulting center-to-center distances of 76 mm defined the pitch. The domain was discretized using a mesh with 2 mm resolution.
(TIF)

**S10 Fig. Probe locations and source geometries.** The full set of simulations and probe locations used in our study. Each panel shows a snapshot of the plumes from the most distal sources, with the two source locations indicated by the white 'o's. All simulations used 16 equally spaced sources, inclusive, between the two distal locations indicated, except for the simulations in panel F, which used 32 sources. The nine probe locations in each simulation are marked with '×'. (**A**) The main simulations used in our study, with sources transverse to the

direction of the flow. The principal probe location that we discuss in the Main Text is at the blue '×'. (**B**) Simulations with sources at 45 degrees to the flow. (**C**) Simulations with sources parallel to the flow. (**D**) The principal supplementary simulations we analyze in the main text, with sources transverse to the flow. The principal probe location that we analyze is at the blue '×'. (**E**) Supplementary simulations with sources at 45 degrees to the flow. (**F**) Supplementary simulations with sources parallel to the flow. This set of simulations used 32 sources, rather than 16.
(TIF)

**S11 Fig. Example plumes and concentration profiles for the supplementary set of simulations.** Compare to Fig 3.
(TIF)

**S12 Fig. Frequency decomposition of correlations for the second set of simulations, using a 1s Hann window.** Compare to Fig 9.
(TIF)

**S13 Fig. Fisher information vs. frequency for the data in the second set of simulations, using a 1s Hann window.** Compare to Fig 12. The fits are at for intersource distances of 0.1, 0.4 and 0.7 $\phi$. The intersource distances for the supplementary data start at lower pitch values than for the data in the Main Text because the pitch for the supplementary simulations is $\sim 3\times$ larger.
(TIF)

**S14 Fig. Amplitude spectra of odour concentration profiles from all odour sources measured at the probe location, for the two simulated flows and the three surrogate datasets used in the text.** Discrete Fourier transforms were computed for consecutive 1-second windows that overlapped by 500 msec, amplitudes were averaged and scaled to have the same value at 1 Hz. Surrogate datasets are indexed by their information content ('all=': all frequencies equally informative; 'high>low': high frequencies more informative than low frequencies). The surrogate datasets (all =) and (high > low) were used in Fig 11 panels B, and C, respectively.
(TIF)

**S15 Fig. Modeling the distribution of observed correlations.** As in Fig 7 but showing the fits to the 1 Hz data, highlighting the poor fits to the data.
(TIF)

**S16 Fig. Modeling the distribution of observed correlations.** As in Fig 7 but showing the fits to the 1 Hz data, and when computing all statistics over 2-second Hann windows instead of the 1-second windows used in Fig 7.
(TIF)

**S17 Fig.** Coupling of concentration profiles from two sources at each frequency as a function of intersource separation, expressed in terms of (**A**) phase and (**B**) strength of the coupling. (**C**) Strength (saturation) and phase (hue) together. (**D**) Out-of-phase correlations, computed as $\beta \sin(\theta)$.
(TIF)

**S18 Fig. As in S17 Fig but for sources that are arranged parallel to the flow (see e.g. Fig 13A).**
(TIF)

**S19 Fig. As in S17 Fig but for sources that are arranged at 45 degrees to the flow (see e.g. Fig 13C).**
(TIF)

**S20 Fig. Velocity autocorrelation functions and corresponding integral length scales ($L_U$) for the simulations in the Main Text, evaluated along the midline ($y = 0$) at $x = 0$ (the $x$ location of the odour sources, labeled 'origin') and at the probe location (labeled 'probe').** (**A**) Velocity in the $x$-direction (parallel to the mean flow) autocorrelated along the same direction. There are fewer data points at the 'probe' location since the largest $x$-displacement at that location is to the origin, while the largest $x$-displacement for the 'origin' extends past the probe location. (**B**) Velocity in the $y$-direction (perpendicular to the mean flow), autocorrelated along the same direction. The maximum displacement is approximately half that of panel A because the data in that panel spans the entire width of the simulation domain, while the data in this panel only extends from the midline to the upper and lower boundaries.
(TIF)

**S21 Fig. Elbow plots for other probe locations and source geometries.** As in Fig 13 but using a 2-second Hann window.
(TIF)

**S22 Fig. Elbow plots for other probe locations and source geometries.** As in Fig 13 but using a 0.5-second Hann window.
(TIF)

**S23 Fig. Elbow plots for other probe locations and source geometries.** As in Fig 13 but using a 1-second Kaiser-16 window.
(TIF)

**S24 Fig. Elbow plots for other probe locations and source geometries, for the supplementary dataset.** As in Fig 13 but for the Supplementary simulations.
(TIF)

**S25 Fig. Elbow plots for other probe locations and source geometries.** As in S24 Fig but using a 2-second Hann window.
(TIF)

**S26 Fig. Elbow plots for other probe locations and source geometries.** As in S24 Fig but using a 0.5-second Hann window.
(TIF)

**S27 Fig. Elbow plots for other probe locations and source geometries.** As in S24 Fig but using a 1-second Kaiser-16 window.
(TIF)

## Acknowledgments

We thank the members of the Crimaldi and Schaefer labs, the Neuroscience Interest Group at the Francis Crick Institute, the Latham lab at the Gatsby Computational Neuroscience Unit, and the members of Odor2Action IRG3 for their feedback. We also thank Jonathan Victor for his detailed comments on this work, in particular for pointing out the need to consider the influence of out-of-phase correlations, and for helping to clarify and simplify our model assumptions.

## Author Contributions

**Conceptualization:** Sina Tootoonian, Aaron C. True, John P. Crimaldi, Andreas T. Schaefer.

**Data curation:** Sina Tootoonian, Aaron C. True, Elle Stark.

**Formal analysis:** Sina Tootoonian, Aaron C. True, Elle Stark.

**Funding acquisition:** John P. Crimaldi, Andreas T. Schaefer.

**Investigation:** Sina Tootoonian, Aaron C. True, Elle Stark, John P. Crimaldi.

**Methodology:** Sina Tootoonian, Aaron C. True.

**Resources:** John P. Crimaldi, Andreas T. Schaefer.

**Software:** Sina Tootoonian, Aaron C. True, Elle Stark.

**Visualization:** Elle Stark.

**Writing – original draft:** Sina Tootoonian, Aaron C. True, Elle Stark, John P. Crimaldi, Andreas T. Schaefer.

**Writing – review & editing:** Sina Tootoonian, Aaron C. True, Elle Stark, John P. Crimaldi, Andreas T. Schaefer.

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
