## [Decision Letter · Decision Letter 0]

4 Feb 2024

PONE-D-23-38285Quantifying spectral information about source separation in multisource odour plumesPLOS ONE

Dear Dr. Tootoonian,

Thank you for submitting your manuscript to PLOS ONE. After careful consideration, we feel that it has merit but does not fully meet PLOS ONE’s publication criteria as it currently stands. Therefore, we invite you to submit a revised version of the manuscript that addresses the points raised during the review process.

We look forward to receiving your revised manuscript.

Kind regards,

Roi Gurka

Academic Editor

PLOS ONE

Journal Requirements:

3. To comply with PLOS ONE submissions requirements, please provide the following information in the Methods section of the manuscript and in the “Ethics Statement” field of the submission form (via “Edit Submission”):  

*  Please indicate whether an animal research ethics committee prospectively approved this research or granted a formal waiver of ethics approval.*  Please enter the name of your Institutional Animal Care and Use Committee (IACUC) or other relevant ethics board. Also include an approval number if one was obtained.

*   If anesthesia, euthanasia, or any kind of animal sacrifice is part of the study, please include briefly in your statement which substances and/or methods were applied.

For additional information about PLOS ONE submissions requirements for ethics oversight of animal work, please refer to http://journals.plos.org/plosone/s/submission-guidelines#loc-animal-research  

6. We note you have included a table to which you do not refer in the text of your manuscript. Please ensure that you refer to Table 2 in your text; if accepted, production will need this reference to link the reader to the Table.

Reviewers' comments:

Reviewer's Responses to Questions

**Comments to the Author**

1. Is the manuscript technically sound, and do the data support the conclusions?

Reviewer #1: Yes

Reviewer #2: Yes

2. Has the statistical analysis been performed appropriately and rigorously? 

Reviewer #1: Yes

Reviewer #2: Yes

3. Have the authors made all data underlying the findings in their manuscript fully available?

Reviewer #1: Yes

Reviewer #2: Yes

4. Is the manuscript presented in an intelligible fashion and written in standard English?

Reviewer #1: Yes

Reviewer #2: Yes

5. Review Comments to the Author

Reviewer #1: This paper investigates the potential of high-frequency olfactory acuity recently observed in mice (see Ref. 36) for spatial localization of odor sources. The authors use computational fluid dynamics simulations to generate complex odor concentration fields in a 2D environment, and then study the temporal correlations between the odor concentrations generated from two spatially separated sources. More specifically, their main aim is to show how informative these correlations can be concerning the distance between the two odour sources.

To this end, they fix a point in space where they measure the temporal correlations between the two signals and then, using simplifying assumptions, derive some analytic expressions for the Fisher information as a way of quantifying the spatial information about source separation contained in each spectral component of such correlations.

Their results show that high frequencies are more informative than low frequencies when sources are close relative to the flow's large eddies, and vice versa.

These findings suggest that the murine olfactory system's high-frequency sensitivity may indeed enable precise estimation about the intersource distance. Moreover, the authors here introduce an approach to quantify the information content of the time signal of odour concentration emitted by a source in a turbulent environment.

The work is well-written and both the numerical and analytical methods are thoroughly presented to the reader. The data shown support the authors' main statements and I am sure this work warrants publication in some form. However, before recommending it for publication in this journal, I would like the authors to address some technical concerns and implement a few minor changes that in my view would improve the manuscript readability and impact.

Major points:

1) Firstly, I am a bit worried about the generality of their statements. Their analysis appears to be focused on one quite specific setup in which:

- the sources are separated only in the direction orthogonal to the mean wind. What happens if they are instead shifted in the direction parallel to it (i.e. one behind the other)? Does their main argument still hold (high frequencies being more informative as sources are closer)? This is not obvious to me. Then, depending on the outcome of such analysis, I would recommend to at least comment in the discussion on the general case in which the displacement vector between the two sources forms an angle 0<θ<π/2.

- the correlations are measured only in one point in space. How does the Fisher information about the intersource separation change with the distance from the sources? Is there any qualitative change in the colormaps in Fig. 11 when shifting the point in either the crosswind or downwind direction?

I think both these points are quite important as animals typically move while looking for an odour source, and therefore, in order for the results presented here to be valuable, I believe they must prove robust to such changes in the setup.

2) The time window used to compute the correlations seems crucial for any conclusions to be drawn. A natural question would then be how this effective time window of "observation" is reflected in biological systems. Is there a typical one that can be considered for animals/insects? If so, how does this compare with the values used in their numerical simulations? The authors quickly mention this connection at lines 375-381 in the discussion, but I would at the very least ask them to comment further on this point by addressing more precisely my question and citing the relevant literature.

Minor points:

1) While reading Sec. 2.2.1, I was confused by the fact that, on the one hand, the authors try to improve the model of correlations derived in the previous section. On the other hand, to the best of my understanding, they later on (Sec. 2.3) forget about this corrections and still use the most simplified model to generate all the analytic results thereafter. At this point, if I am not mistaken, why not moving this section with the improved models in the Supplementary Material and include some of the calculations leading to the final expression of the Fisher information in the main body instead? I think this would improve the readabiity of this technical (and very important) part of the paper.

2) I would recommend the authors to make more precise citations in most of the sentences in the introduction and avoid putting references in bulks. Note, for e.g., how the first sentence does not reference any paper in the literature while bringing up important points. In that regard, I would also suggest to expand the introductory paragraph about olfactory search in complex or turbulent environments*. I am sure this would definitely strengthen the problem statement while increasing the visibility of this research.

3) Can the results presented here be useful as a basis to implement optimal navigation strategies in complex landscapes? I am thinking of active agents across all scales (from microswimmers to moths, mice, birds or even robots). In my opinion, it could be a nice potential application that can be inserted as a further outlook in the discussion section**.

* Suggested relevant literature for olfactory search:

- C. David et al., Nature, 303:804–806 (1983).

- M Durve et al., Physical Review E 102 (1), 012402 (2020)

- G. Reddy et al., Annual Review of Condensed Matter Physics 13 (2022)

** Suggested relevant literature for optimal navigation:

- J. Pinti et al., Theoretical Ecology 13 (4), 583-593 (2020)

- L. Piro et al., New Journal of Physics 24 (9), 093037 (2022)

- L. Piro et al., Frontiers in Physics 10, 1125 (2022)

- J. Jiang et al., Advanced Intelligent Systems, 4(5):2100279 (2022)

- R. A. Heinonen et al., Physical Review E 107 (5), 055105 (2023)

Reviewer #2: The manuscript presented by Tootoonian and colleagues is technically strong and develops an interesting approach for the analysis of odor plumes. The main weakness I would like to see addressed is a seemingly insufficient explanation and justification of the underlying assumptions of the study.

Firstly, assumptions of Gaussian profiles in Assumptions 1 and 4 could be justified when it is first mentioned by referring to Figures 5 and 6. And those plots could be further analyzed to explicitly check the Gaussian nature, by standardizing and then plotting on a log scale to see if the log density falls quadratically.

Assumption 2 in particular, and how spatial dependence in general is handled in this work, is presented in a rather confusing manner. It seems that the ideas and words “source”, “source separation” and “location” are used somewhat interchangeably, when they should be treated separately. In general, if these results are expected to hold in different locations of the plume relative to the source locations then this should explored and justified more, eg. by analyzing the distribution of the Fourier coefficients or r_n at different locations in the plume. Otherwise, the authors should make clear that their findings are limited to samples taken at a single location. It may be that things become much more complicated off-center, but at least if a few samples are taken at different, significantly spaced, downwind locations along the centerline and similarly analyzed, this will make the work more general.

Additionally, it is not clear to me how the assumption of the form of rho_n in equation 18 is linked to the assumption in equation 33. Justification of exponential dependence, eg. in Fig 8 could also be more carefully demonstrated by plotting in log-scale.

Minor comments:

• Line 14, “correlation of odor concentration fields” is not very clear. Maybe “correlation of odor concentration timeseries”

• Typos in lines 88 and line 196, and line 235

• Instead of denoting as information between n and s, might be better to denote as information between r_n and s, but I’m happy to let the authors decide

• The purpose of the analysis of the surrogate data could be made more clear. Maybe a little more in the main text about how it is generated will shed more light on generally what makes the spectral distribution of information flat vs. non-flat.

• The motivation behind the study and ethological relevance of the findings could still be made stronger. Perhaps further expansion in the introduction of the findings in citation [35] and how resolving source separations at such at this scale can matter in real contexts around line 361.

6. PLOS authors have the option to publish the peer review history of their article (what does this mean?). If published, this will include your full peer review and any attached files.

Reviewer #1: No

Reviewer #2: No

---

## [Author Response · Author response to Decision Letter 0]

6 Aug 2024

We thank the reviewers for the detailed feedback. We believe that we have now addressed most of their concerns, as indicated in detail in the attached point-by-point rebuttal.

---

## [Decision Letter · Decision Letter 1]

28 Aug 2024

Quantifying spectral information about source separation in multisource odour plumes

PONE-D-23-38285R1

Dear Dr. Tootoonian,

We’re pleased to inform you that your manuscript has been judged scientifically suitable for publication and will be formally accepted for publication once it meets all outstanding technical requirements.

Kind regards,

Roi Gurka

Academic Editor

PLOS ONE

Additional Editor Comments (optional):

Please address reviewer #2 comments and revise the manuscript accordingly before submitting the final version.

Reviewers' comments:

Reviewer's Responses to Questions

**Comments to the Author**

1. If the authors have adequately addressed your comments raised in a previous round of review and you feel that this manuscript is now acceptable for publication, you may indicate that here to bypass the “Comments to the Author” section, enter your conflict of interest statement in the “Confidential to Editor” section, and submit your "Accept" recommendation.

Reviewer #1: All comments have been addressed

Reviewer #2: All comments have been addressed

2. Is the manuscript technically sound, and do the data support the conclusions?

Reviewer #1: Yes

Reviewer #2: Yes

3. Has the statistical analysis been performed appropriately and rigorously? 

Reviewer #1: Yes

Reviewer #2: Yes

4. Have the authors made all data underlying the findings in their manuscript fully available?

Reviewer #1: Yes

Reviewer #2: Yes

5. Is the manuscript presented in an intelligible fashion and written in standard English?

Reviewer #1: Yes

Reviewer #2: Yes

6. Review Comments to the Author

Reviewer #1: I thank the authors for their thorogh revision of the paper and response. I believe they have convincingly addressed all my concerns and the paper has now greatly improved in readability and potential impact. Therefore, I recommend it for publication in its current form.

Reviewer #2: The authors have addressed all comments and the work is now ready for publication. I commend them for their efforts in addressing these comments for their rigorous and detailed study. Having said that, there are a few minor revisions that should still be made. References to supplementary figure numbers often seem to be off by one. There are also several typos still present, eg. line 310, line 1225, the subtitle around line 1200.

Then just as other comments which the authors are free to discard-in the discussion, the authors may also want to mention the fact that even at the earliest stages of olfactory signal processing in animals, there is adaptation and nonlinearities. How this might affect the utility of different spectral components might be an interesting avenue for future research. Additionally, in reality odor identities are encoded as activity patterns across different receptors, and the same odorant can activate multiple receptors. Thus, from two different sources one may get additional correlations in sensory neuron activity due to receptor promiscuity, in addition to the correlations due to the turbulent effects presented by the authors. The interplay between these effects could also be an interesting avenue for further research.

7. PLOS authors have the option to publish the peer review history of their article (what does this mean?). If published, this will include your full peer review and any attached files.

Reviewer #1: No

Reviewer #2: No

---

## [Editor Report · Acceptance letter]

8 Nov 2024

PONE-D-23-38285R1 

PLOS ONE

Dear Dr. Tootoonian, 

I'm pleased to inform you that your manuscript has been deemed suitable for publication in PLOS ONE. Congratulations! Your manuscript is now being handed over to our production team.

Kind regards, 

on behalf of

Dr. Roi Gurka 

Academic Editor

PLOS ONE